# How norm violators rise and fall in the eyes of others: The role of sanctions

**Florian Wanders**[1], **Astrid C. Homan**[1], **Annelies E. M. van Vianen**[1]*, **Rima-Maria Rahal**[2], **Gerben A. van Kleef**[3]

1 Work and Organizational Psychology, University of Amsterdam, Amsterdam, The Netherlands, 2 Social Psychology, Tilburg University, Tilburg, The Netherlands, 3 Social Psychology, University of Amsterdam, Amsterdam, The Netherlands

* a.e.m.vanvianen@uva.nl

**Data Availability Statement:** Data and analysis code are available through https://osf.io/xjpe5/.

**Funding:** The author(s) received no specific funding for this work.

## Abstract

Norm violators demonstrate that they can behave as they wish, which makes them appear powerful. Potentially, this is the beginning of a self-reinforcing loop, in which greater perceived power invites further norm violations. Here we investigate the possibility that sanctions can break this loop by reducing the power that observers attribute to norm violators. Despite an abundance of research on the effects of sanctions as deterrents for norm-violating behavior, little is known about how sanctions may change perceptions of individuals who do (or do not) violate norms. Replicating previous research, we found in two studies ($N_1$ = 203, $N_2$ = 132) that norm violators are perceived as having greater volitional capacity compared to norm abiders. Qualifying previous research, however, we demonstrate that perceptions of volition only translate into attributions of greater power in the absence of sanctions. We discuss implications for social hierarchies and point out avenues for further research on the social dynamics of power.

## Introduction

Social norms—implicit or explicit rules or principles that are understood by members of a group and that guide and/or constrain behavior [1]–create a shared understanding of what is acceptable within a given context and thereby contribute to the functioning of social collectives [2–4]. Accordingly, research has documented that people who violate norms tend to elicit negative responses in others, including unfavorable social perceptions [5], negative emotions [6–8], scolding [9], gossip [10], and punishment [11–13]. Intriguingly, however, research has also demonstrated that norm violators are perceived as powerful [5], high in status [14], and influential [15]. This possibly opens the door to a "self-reinforcing loop" (p. 351 [16]): Norm violators appear powerful and bystanders may submit to powerful others [17], thereby inviting further norm violations and consolidating norm violators' power [5]. The question then arises: How can we prevent people from gaining unjustified influence through norm violations? Here we investigate whether sanctions reduce the extent to which norm violators appear powerful, thereby breaking the self-reinforcing loop to power that norm violations can set off.

**Competing interests:** The authors have declared that no competing interests exist.

## Norm violations signal power

We define norm violations as any behavior that infringes on a norm [5], whether informal (i.e., learned by observing others) or formal (i.e., written). Norm violations are ubiquitous, from talking at the movies to using public transport without a ticket. These behaviors violate social norms that are both endorsed and enacted by most members of a group (injunctive and descriptive norms, respectively) [4]. Injunctive and descriptive norms are individually perceived but when people are cognizant of prevailing norms and endorse these norms, both types of norms can converge and be shared at the collective level [18]. By ignoring the norms that bind others, norm violators demonstrate that they can act as they wish and do not fear interference from others [5]. This is a freedom that typically comes with higher rank [19].

The influential approach/inhibition theory of power [20] states that power, which is commonly defined as asymmetrical control over valuable resources that enables influence, liberates behavior, whereas powerlessness constrains it. Indeed, ample research supports that power renders people more likely to act, even if the resulting behavior is inappropriate or harmful [21, 22]. Because behavioral freedom is thus intimately associated with power, people who observe unchecked behavior of others may make inferences about others' level of power. Indeed, people who act as they wish and disregard social norms are perceived as having high status [14], influence [15], and power [5]. Furthermore, these perceptions can, under particular circumstances, fuel actual granting of power, for instance via the conferral of control over outcomes, voting, and leadership endorsement [23, 24].

In line with the notion that power liberates behavior, previous research has demonstrated that norm violators are perceived as powerful because they appear to experience the freedom to act as they please [5, 14, 15]–that is, they are high on *volitional* capacity. In other words, norm violators are perceived as powerful because their behavior signals an underlying quality, namely the freedom to act at will. This argument resonates with costly signaling theory [25, 26], which states that any seemingly costly behavior (involving large investments or risks of receiving negative outcomes) functions as a signal of an underlying characteristic [25, 26]. An example of costly behavior is the reckless driving of young men as to show their strength and skills to peers and potential mates, risking serious injury or death—a type of behavior that is under particular circumstances "rewarded" with power [27]. Norm violations are potentially costly as they are frequently sanctioned [14] by means of formal (e.g., legal) punishment [28] and/or informal (social) punishment (e.g., anger, social exclusion [29, 30]). According to costly signaling theory, people who engage in potentially costly norm-violating behavior signal that they possess traits that allow them not to worry about interferences from others. Because this capacity to do what one wants is typically reserved for the powerful [31], norm violators appear powerful when there are no additional cues that provide direct information about this attribute [5].

## Sanctions curb norm violators' perceived power

If norm violations signal power, this opens the door to a self-reinforcing loop [5, 16]. Norm violators' claim to power is likely to be granted because people tend to submit to powerful others [17, 24, 32]. For example, people who interrupt others during meetings may be granted influence by receiving more time to speak [14, 33]. As a consequence, their contributions may be noted more readily, which increases their chances for influence and promotion [34]. Norm violators may therefore climb up in social hierarchies. The question then arises: Can people be prevented from gaining power through norm violations?

Here we adopt a social-perceptual lens and investigate whether sanctions reduce the extent to which norm violators appear powerful. Specifically, we propose that sanctions reduce the

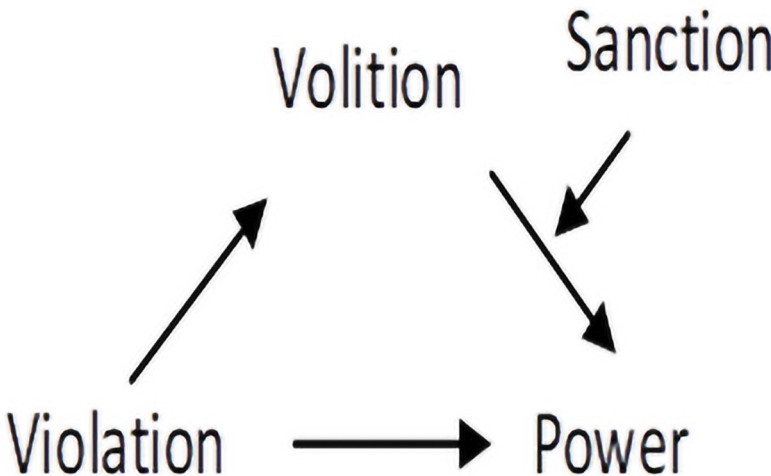

**Fig 1. Conceptual model of the proposed moderating effect of sanctioning.**

signal of power that norm violators' apparent volitional capacity sends. Bystanders easily infer that norm violators are free to act according to their own volition [5]. In the absence of additional information, this inference of volitional capacity functions as a signal of power [5, 16]. However, we argue that if bystanders receive information that norm violators are sanctioned, they no longer need to rely on such signals. That is, they may directly conclude that norm violators who are reprimanded for their behavior do not have the power they seemed to have but are bound by the same norms that bind others around them. To summarize, we argue that bystanders perceive norm violators as powerful because they infer that norm violators have the capacity to act according to their own volition (replication of Van Kleef et al [5]). However, we propose that sanctions reduce the extent to which observers perceive norm violators as powerful by severing the link between volition and power perceptions (see Fig 1).

## Overview

The goal of the current manuscript was to investigate whether sanctions reduce the extent to which norm violators are seen as powerful. In our experimental design we focus on the violation of a legal norm that most people in society tend to endorse and enact, and sanctions refer to formal rather than informal sanctions. We present the results of two studies which replicate the finding that unsanctioned norm violators appear powerful [5], and support the current hypothesis that sanctions curb the effect of norm violations on power perceptions. The investigation of the exact mechanism underlying this effect was in part exploratory, and we denote where this was the case when presenting our results.

## Study 1

### Methods

**Participants and design.** Study 1 employed a 2 (norm violation: abide vs. violate) × 2 (sanctioning: no sanction vs. sanction) between-subjects design, and participants could win one of four 15€ vouchers. This study was part of a student project using a cell size of about 50 participants and including an additional exploratory condition ($n = 121$) which we do not report here (see S1 File for further information). Ethics approval was obtained from the ethical review board, Faculty of Social and Behavioral Sciences, University of Amsterdam, the

Netherlands (ref.: 2014-WOP-3498). The code of conduct of the German Society for Psychology does not require special permits for international researchers and, for ethical considerations in research, the same codified ethical guidelines apply in Germany as in the Netherlands. All participants provided written informed consent prior to their participation (online, by clicking "yes").

We recruited 236 participants at a German university campus and through social media, of which 203 were retained for analyses (153 women, 50 men, $M_{age}$ = 23.78, range = 18–59). Seventeen participants were removed because they did not complete the questionnaire, and 16 participants were excluded because they failed attention checks. These exclusion criteria were decided a-priori. A sensitivity analysis conducted in G-power suggested that when testing a moderated mediation model involving 5 predictors (norm violation, volition, sanctioning, norm violation x sanctioning, volition x sanctioning) and α = 0.05 the analysis would have a power of 0.80 to detect a small to medium effect ($f^2$ = 0.06). In addition, we calculated ν-statistics [35] for the central tests of our moderated mediation model. The ν-statistic for the regressions of volition on norm violation was ν = 0.897. The ν-statistic for the regression of power on volition, norm violation, sanctioning, and the two-way interactions between violation and sanctioning as well as volition and sanctioning was ν = 1.000. These statistics show that this study was sufficiently powered.

**Manipulation.**  Participants read about a traveler who either purchased a ticket before boarding a train (norm abider) or purchased a snack instead and did not purchase a ticket (norm violator). The norm abider could not find the ticket when approached by a controller on the train but told the controller that he did buy one. Likewise, the norm violator told the controller that he did buy a ticket but said that he had already been checked. The controller then either did not insist on seeing the ticket (no sanction) or did insist and fined the traveler who was unable to show the ticket (sanction; see the S1 File for the full scenarios). Assignment to conditions was random.

**Measures.**  After reading about the traveler, participants indicated to what extent they thought the traveler acted out of his own volition, and to what extent they perceived the traveler as powerful. The measures including all items can be found in the supplementary material. Participants answered a set of additional questions (administered as part of a thesis project) before completing manipulation and attention checks.

*Volition perceptions.* Perceptions of volition (α = .88) were measured with six items [1]. An example item is: "To what extent does this person feel free to do what s/he wants?" with scales ranging from 1 = *not very much*, to 7 = *very much*.

*Power perceptions.* Perceptions of power (α = .88) were measured with a validated 8-item sense of power scale [36]. Example items are: "I think this person has a great deal of power" and "I think this person's wishes do not carry much weight (reverse scored)" with scale anchors ranging from 1 = *strongly disagree*, to 7 = *strongly agree*.

*Manipulation checks.* Two questions each assessed in how far participants thought the traveler violated norms ("To what extent did the traveler violate norms?"; "To what extent did the traveler abide by norms?" [reverse-scored]; *r* = .87) and in how far participants thought the traveler was sanctioned ("To what extent was the traveler sanctioned?"; "To what extent did the traveler get away unsanctioned?" [reverse-scored]; *r* = .93). Scale anchors for both manipulation checks were 1 = *not at all*, and 7 = *extremely*.

*Attention checks.* Participants answered two questions each asking whether traveling without a ticket was allowed/prohibited, whether the traveler did/did not buy a ticket, whether the traveler was/was not fined, and whether the traveler was/was not honest. Answer options were yes versus no, and participants who provided incorrect responses were excluded from the analyses.

## Results

**Manipulation checks.** To test whether the manipulations of norm violation and sanctioning were successful, we ran two separate ANOVAs with norm violation and sanctioning as between-subjects factors. First, the ANOVA with the norm violation manipulation check as dependent variable revealed the expected main effect of norm violation, $F(1,199) = 1085.56$, $p < .001$, $\eta_p^2 = .845$. Norm violators ($M = 6.29$, $SD = 0.78$, 95% CI [6.136, 6.442]) were seen as violating norms to a considerably greater extent than norm abiders ($M = 2.14$, $SD = 1.13$, 95% CI [1.916, 2.361]). Unexpectedly, there was also a main effect of sanctioning, $F(1,199) = 18.41$, $p < .001$, $\eta_p^2 = 0.085$, and a significant interaction effect, $F(1,199) = 18.52$, $p < .001$, $\eta_p^2 = 0.085$. Further probing using simple slopes analysis revealed no significant effect of sanctioning for norm violators, $t(199) = 0.01$, $p = .993$, 95% CI [-0.348, 0.351], $d = 0.002$, but there was a significant effect for norm abiders, $t(199) = 6.06$, $p < .001$, 95% CI [0.728, 1.429], $d = 1.207$: Non-sanctioned norm abiders were perceived as violating norms to a greater extent than sanctioned norm abiders. Given that the effect sizes of the unexpected effects (both $\eta_p^2 = 0.085$) were ten times smaller than that of the intended effect ($\eta_p^2 = .845$) we consider this manipulation successful.

Second, the ANOVA with the sanctioning manipulation check as dependent variable revealed only the expected main effect of sanctioning, $F(1,199) = 1589.38$, $p < .001$, $\eta_p^2 = .889$. Sanctioned travelers ($M = 5.94$, $SD = 1.06$, 95% CI [5.733, 6.149]) were seen as considerably more sanctioned than non-sanctioned travelers ($M = 1.29$, $SD = 0.51$, 95% CI [1.186, 1.388]). Neither the effect of norm violation nor the interaction between sanctioning and norm violation were significant ($F < 2.85$, $p > .093$). Thus, the manipulation was successful.

**Replication of the norm violation-perceived power link.** Next, we aimed to replicate Van Kleef et al.'s [5] norm violation → volition → perceived power links in the absence of sanctions, before investigating how these links are affected by the presence of sanctions. As illustrated in the left-hand panel of Fig 2, a planned contrast revealed that in the absence of sanctions norm violators appeared more powerful than norm abiders, $t(99) = 2.02$, $p = .047$, 95% CI [0.005, 0.690], $d = 0.401$ (see Table 1 for means and standard deviations).

For testing our directional prediction that volition mediates the link between norm violation and perceived power, we used one-tailed tests [37]. Norm violators were seen as acting more according to their own volition compared to norm abiders, $B = 1.65$, $SE = 0.19$, $t(99) = 8.76$, $p < .001$, 95% CI [1.276, 2.023], and greater perceived volitional capacity was, in turn, related to greater perceived power, $B = 0.18$, $SE = 0.09$, $t(98) = 1.99$, $p = .025$, 95% CI [0.029, Inf].

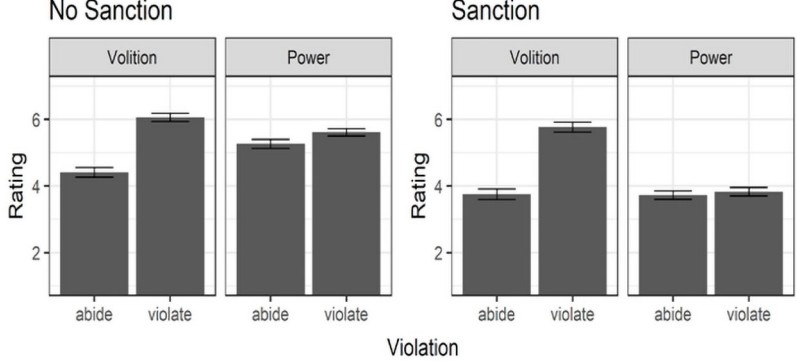

**Fig 2. Means and standard errors for the effects of norm abidance vs. violation on perceptions of volition and power in Study 1 in the absence (left panel) vs. presence (right panel) of sanctions.**

**Table 1. Means and standard deviations for the effects of norm abidance vs. violation on perceptions of volition and power in Study 1 in the absence vs. presence of sanctions.**

| Sanction Violation | No sanction | | Sanction | |
|---|---|---|---|---|
| | **Abide** | **Violate** | **Abide** | **Violate** |
| Volition | 4.41 (1.01) a | 6.06 (0.88) b | 3.75 (1.16) c | 5.77 (1.07) b |
| Power | 5.27 (0.94) a | 5.61 (0.79) b | 3.73 (0.90) c | 3.83 (0.89) c |

Note. Means within a row with a different subscript differ at $p < .05$.

Bootstrapped confidence intervals indicate that the indirect effect of norm violation on perceived power via volition was significant, $B_{indirect} = 0.30$, $SE = 0.15$, 95% CI [0.010, 0.582], $\upsilon = 0.029$. The effect size $\upsilon$ indicates a sufficient although small indirect effect [38]. We therefore consider the replication of the norm violation → volition → perceived power links successful.

**The role of sanctioning.**   Concerning the effect of sanctioning on the norm violation → volition → perceived power link, we predicted that sanctioning would reduce the extent to which norm violators appear powerful. Furthermore, we proposed that sanctioning would reduce the signal of power that norm violators' apparent volitional capacity sends. We tested this idea in three steps. First, we tested whether sanctioning reduced the extent to which norm violators were seen as powerful. A planned contrast suggests that sanctioned norm violators were indeed perceived as less powerful than non-sanctioned norm violators $t(100) = -10.68$, $p < .001$, 95% CI [-2.114, -1.452], $d = -2.115$ (see Table 1 for means and standard deviations).

Next, we explored where in the norm violation → volition → perceived power links sanctions exerted their moderating impact. Our theoretical argument suggested that observers perceive norm violators as having greater volitional capacity than norm abiders regardless of whether they are sanctioned, whereas they will perceive norm violators as powerful only if they are not sanctioned. In line with this idea, a mixed-model ANOVA among norm violators with sanctioning (no sanction vs. sanction) as between-subjects factor and scale (volition vs. power) as within-subjects factor revealed—besides significant main effects of sanctioning, $F(1,100) = 51.23$, $p < .001$, $\eta_p^2 = .339$ and scale $F(1,100) = 122.81$, $p < .001$, $\eta_p^2 = .551$—a significant interaction between both, $F(1,100) = 47.70$, $p < .001$, $\eta_p^2 = .323$. As Fig 3 shows, whereas sanctions did not significantly affect the extent to which norm violators appeared to act according to their own volition, $t(100) = -1.50$, $p = .136$, 95% CI [-0.675, 0.093], $d = -0.298$, they significantly reduced perceptions of power $t(100) = -10.68$, $p < .001$, 95% CI [-2.114, -1.452], $d = -2.115$. This suggests that sanctions reduce the signal of power that norm violators' apparent volitional capacity sends.

In a final step, we tested whether sanctioning moderated the effect of volition on perceived power in the norm violation → volition → perceived power link. Sanctioning moderated the effect of volition on perceived power in the mediation chain when the confidence interval for the product $a \times b_2$ of the effect of norm violation on volition (a in Fig 4, left panel) and the interaction of volition and sanction on power perception ($b_2$) excludes zero [39]. See the supplement for a detailed explanation.

Contrary to our expectations, this was not the case, $B = 0.10$, $SE = 0.2$, 95% CI [-0.294, 0.520]. Whereas the effect of norm violation on volition (a) was significant, $B = 1.85$, $SE = 0.15$, $t(201) = 12.40$, $p < .001$, 95% CI [1.553, 2.141], the interaction between volition and sanctioning on power ($b_2$) was not, $B = 0.05$, $SE = 0.12$, $t(197) = 0.45$, $p = .651$, 95% CI [-0.181, 0.289], rendering the product $a \times b_2$ nonsignificant. We therefore cannot conclude that sanctioning reduced the extent to which norm violators' apparent volitional capacity translated into power perceptions. Fig 4 (right panel) illustrates this absence of an interaction between sanctions and

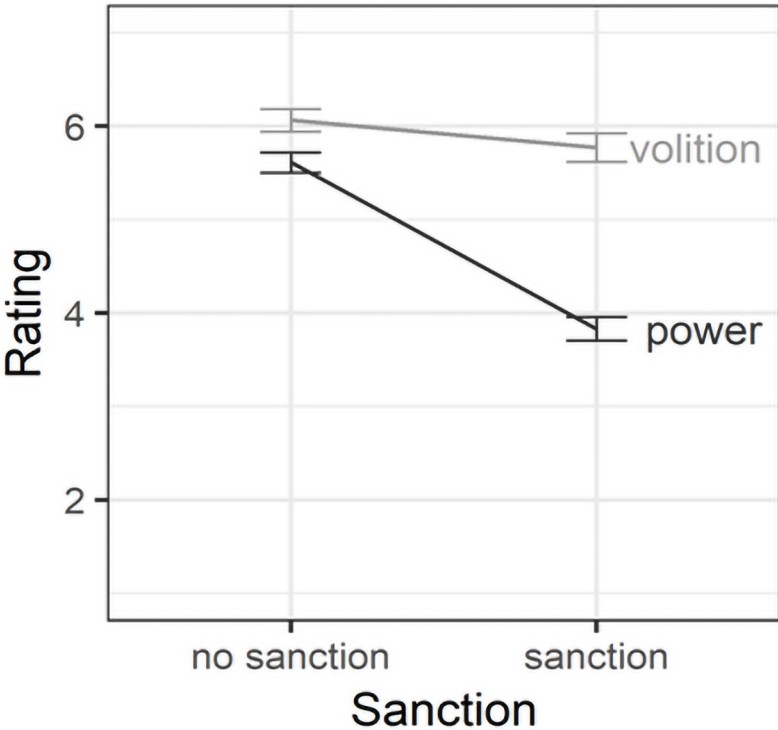

**Fig 3. Effects of sanctions on perceptions of norm violators in Study 1.** Error bars are standard errors around the mean.

volition (slopes are similar across conditions) and shows that sanctioning directly reduced perceptions of power.

## Discussion

Study 1 replicated the finding that norm violators are seen as acting more according to their own volition than norm abiders, and that greater volition in turn related to greater inferences

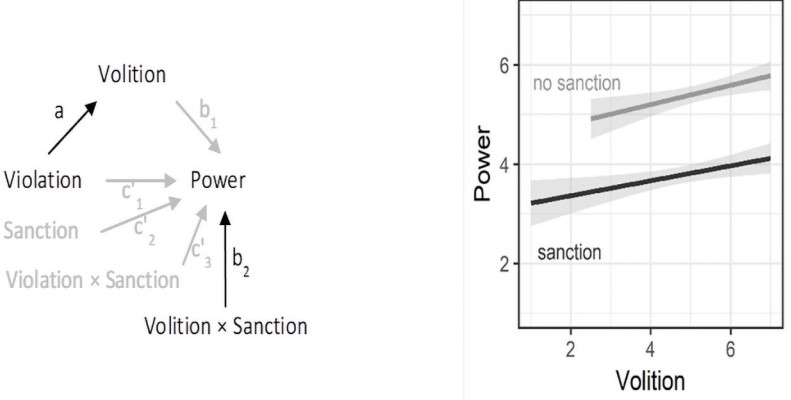

**Fig 4. Statistical model (left) of the proposed moderating effect of sanctioning in Study 1.** Black arrows in the statistical model highlight relevant effects for moderated mediation ($ab_2$). Simple slopes with standard errors (right) illustrate $b_2$, the lack of an interaction of volition and sanctions on power perceptions.

of power [5]. As expected, sanctioning reduced the extent to which norm violators were seen as powerful. However, sanctioning did not significantly affect the extent to which norm violators appeared to act according to their own volition. Although this is consistent with our theoretical model, which proposes that sanctioning targets the power-signaling effect of volition in the norm violation → volition → perceived power mediation chain, we found no full support for this pattern. Instead, sanctioning directly reduced perceptions of power irrespective of volition. One explanation for why sanctioning did not moderate the power-signaling effect of volition could be that volition was not strongly linked to power perceptions in this study in the first place. Therefore, we aimed to replicate the norm violation → volition → perceived power chain in a second study which also allowed us to improve the ecological validity of our design.

## Study 2

The 2(violate vs. abide) × 2(no sanction vs. sanction) design of Study 1 allowed us to test our predictions in a single moderated mediation model. Yet, despite its elegance, this design necessitated a compromise: To enable orthogonal manipulations of norm violation and sanctioning, neither the norm violator (who never purchased a ticket) nor the norm abider (who lost it) showed a valid ticket, which is sanctionable behavior. Although this enabled a full-factorial design allowing different comparisons between conditions, including a condition with sanctions for a norm abider who lost the ticket, may have undermined the credibility of the scenario, and renders interpretation of the results less straightforward. First, norm violators might have appeared more powerful than norm abiders not because norm violators demonstrated volitional capacity, but because norm abiders seemed incapable. Second, one might question whether norm abiders who lost their ticket really abided by norms, as, according to German train regulations, travelers must at all times be able to show a valid ticket. Therefore, in Study 2, we let the norm abider buy and show a ticket to the controller, moving from the 2×2 design of Study 1 to a 3-cell design.

### Methods

**Participants and design.**　Study 2 employed a 3-cell (norm abider vs. norm violator vs. sanctioned norm violator) between-subjects design and relied on a sample of Dutch participants that was collected as part of a larger project. Participants could win one of five 10€ vouchers. Ethics approval was obtained from the ethical review board, Faculty of Social and Behavioral Sciences, University of Amsterdam (ref.: 2017-COP-8050). All participants provided written informed consent prior to their participation (online, by clicking "yes").

To ensure comparable cell sizes as in Study 1, we recruited 159 participants at the university, of which 132 were retained for analyses (83 women, 49 men, $M_{age}$ = 25.80, range = 18–66). Seven participants were removed because they did not complete the questionnaire, and an additional 20 participants were excluded because they failed attention checks. These exclusion criteria were decided a-priori. A sensitivity analysis conducted in G-power suggested that with 5 predictors (experimental condition 1 [non-sanctioned norm violators vs. abiders], experimental condition 2 [non-sanctioned norm violators vs. sanctioned norm violators], volition, violation x condition 1, volition x condition 2) and α = 0.05 the analysis would have a power of 0.80 to detect a small to medium effect ($f^2$ = 0.10). In addition, we calculated ν-statistics [35] to establish sufficient power. The central test in Study 2 constituted the regression of power on the interaction between volition and experimental condition, which resulted in a ν-statistic of ν = .999 (regressing of volition on experimental condition resulted in a ν-statistic of 0.955). This indicates that our study was sufficiently powered.

**Manipulation.** As in Study 1, participants read about a traveler who either purchased a ticket before boarding a train (norm abider) or purchased a snack instead (and no ticket). When approached by a controller, the norm abider showed the ticket. The norm violator told the controller that he did buy a ticket but said that he had already been checked. The controller then either did not insist on seeing the ticket (norm violator) or did insist and fined the traveler who was unable to show the ticket (sanctioned norm violator; see the S1 File for the full scenarios). Assignment to conditions was random.

**Measures.** After reading about the traveler, participants completed the same measures of perceived power ($\alpha$ = .87) and volition ($\alpha$ = .85) as in Study 1. Besides completing manipulation and attention checks (see below), participants answered a set of additional questions as part of a student project, which were not analyzed (see the S1 File).

*Manipulation checks.* Three questions assessed in how far participants thought the norm violator violated norms: "He behaved in line with norms", "He violated norms", and "He behaved appropriately" (reverse coded, $\alpha$ = .92; adapted from Stamkou et al [23]). Three further questions assessed in how far participants thought the traveler was sanctioned: "The traveler was punished", "The traveler had to pay for his behavior", and "The traveler was fined" ($\alpha$ = .96). Scale anchors for all scales in this study ranged from 1 = *completely disagree*, to 7 = *completely agree*.

*Attention check.* Participants were asked whether the traveler bought a ticket and whether the controller fined the traveler. Answer options were yes versus no, and participants who provided incorrect responses were excluded from the analyses.

## Results

Separate ANOVAs on the manipulation checks with experimental condition as between subjects variable revealed significant differences between conditions on both the norm violation manipulation check, $F(2,129) = 161.62$, $p < .001$, $\eta_p^2 = .715$, and the sanctioning manipulation check, $F(2,129) = 179.08$, $p < .001$, $\eta_p^2 = .735$. Participants perceived both the sanctioned ($M = 5.89$, $SD = 0.87$, 95% CI [5.630, 6.143]) and the non-sanctioned norm violator ($M = 5.91$, $SD = 1.03$, 95% CI [5.585, 6.237]) to have violated norms to a greater extent than the norm abider ($M = 2.41$, $SD = 1.23$, 95% CI [2.036, 2.782]). Participants also perceived the sanctioned norm violator ($M = 5.95$, $SD = 0.85$, 95% CI [5.702, 6.199]) as having been sanctioned to a greater extent than either the non-sanctioned norm violator ($M = 2.28$, $SD = 1.13$, 95% CI [1.927, 2.643]), or the norm abider ($M = 2.20$, $SD = 1.24$, 95% CI [1.821, 2.573]). This shows that the manipulations were successful.

As in Study 1, we aimed to replicate Van Kleef et al.'s [5] norm violation → volition → perceived power links in the absence of sanctioning, before investigating how these links are affected by sanctioning. As illustrated in Fig 5, a planned contrast revealed that, in the absence of sanctions, norm violators appeared more powerful than norm abiders, $t(83) = 7.27$, $p < .001$, 95% CI [0.697, 1.222], $d = 1.579$ (see Table 2 for means and standard deviations).

Concerning the mediating role of volition, norm violators were seen as acting more according to their own volition compared to norm abiders, $B = 1.12$, $SE = 0.15$, $t(83) = 7.30$, $p < .001$, 95% CI [0.816, 1.427], and greater volitional capacity was, in turn, related to greater perceived power, $B = 0.29$, $SE = 0.09$, $t(82) = 3.30$, $p = .001$, 95% CI [0.117, 0.471. Bootstrapped confidence intervals showed that the indirect effect of norm violation on perceived power via volition was significant, $B_{indirect} = 0.33$, $SE = 0.12$, 95% CI [0.118, 0.623], $\upsilon = 0.046$. We therefore consider the replication of the norm violation → volition → perceived power chain successful and proceed to investigate how sanctions affect this chain.

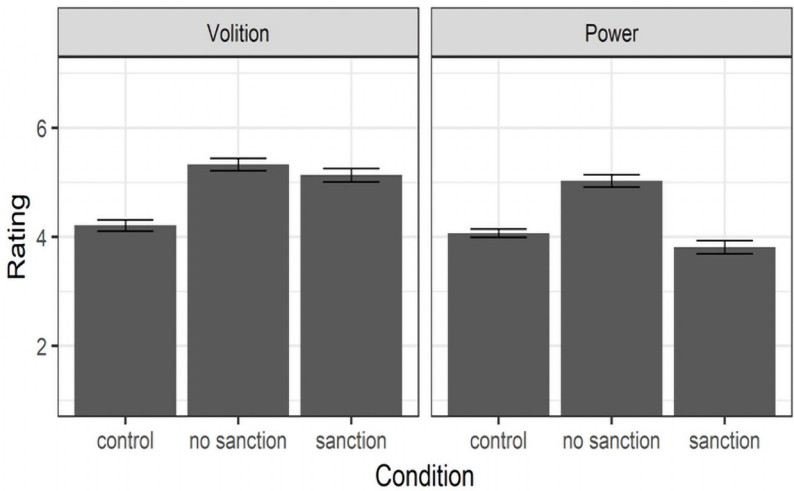

**Fig 5. Means and standard errors for the effects of norm abidance (control), norm violation (no sanction), and sanctioned norm violation (sanction) on inferences of volition and power in Study 2.**

We predicted that sanctioning reduces the extent to which norm violators appear powerful. Furthermore, we proposed that sanctioning reduces the signal of power that norm violators' apparent volitional capacity sends. First, we tested whether sanctioning reduces the extent to which norm violators are seen as powerful. A planned contrast indicates that sanctioned norm violators were indeed perceived as less powerful than non-sanctioned norm violators, $t(86) =$ -7.38, $p < .001$, 95% CI [-1.544, -0.889], $d = $ -1.578 (see Table 2 for means and standard deviations).

Second, mixed-model ANOVA among norm violators with sanctioning (no sanction vs. sanction) as between-subjects factor and scale (volition vs. power) as within-subjects factor revealed—besides significant main effects of sanctioning, $F(1,86) = 28.66$, $p < .001$, $\eta_p^2 = .250$ and scale $F(1,86) = 64.37$, $p < .001$, $\eta_p^2 = .428$—a significant interaction between both, $F(1,86) = 25.30$, $p < .001$, $\eta_p^2 = .227$. As Fig 6 shows, whereas sanctioning did not significantly reduce the extent to which norm violators appeared to act according to their own volition, $t(86) = $ 1.18, $p = .241$, 95% CI [-0.136, 0.533], $d = 0.252$, they significantly reduced perceptions of power $t(86) = 7.38$, $p < .001$, 95% CI [0.889, 1.544], $d = 1.578$. As in Study 1, this is consistent with the possibility that sanctioning reduces the signal of power that norm violators' apparent volitional capacity sends.

In a final step, we tested whether sanctioning moderates the effect of volition on power perceptions. Unlike in Study 1, where the 2×2 design allowed us to test this prediction in a moderated mediation model, we now regressed power perceptions on the interaction between

**Table 2. Means and standard deviations for the effects of norm abidance (control), norm violation (no sanction) and sanctioned norm violation (sanction) on inferences of volition and power in Study 2.**

| Condition | Control | No sanction | Sanction |
|---|---|---|---|
| Volition | 4.21 (0.68) a | 5.33 (0.73) b | 5.13 (0.83) b |
| Power | 4.07 (0.49) a | 5.03 (0.71) b | 3.81 (0.82) a |

*Note.* Means within a row with a different subscript differ at $p < .05$.

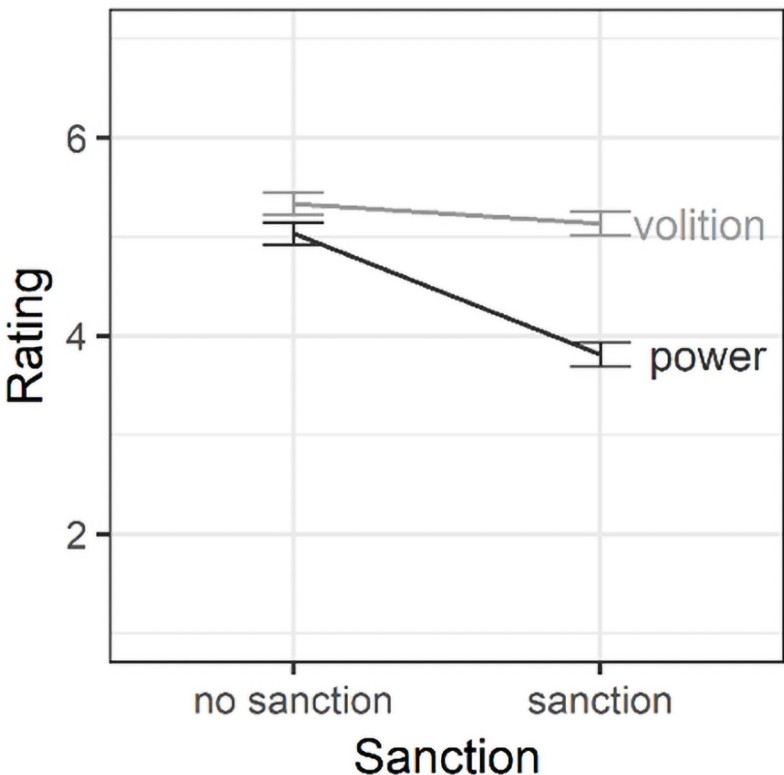

**Fig 6. Effects of sanctioning on perceptions of norm violators in Study 2.** Error bars are standard errors around the mean.

experimental condition and volition, overall $R_{adj}^2 = 0.401$. This analysis corresponds to testing for moderated mediation in Study 1 (specifically, to the $b_2$ path in Fig 4). If sanctioning indeed reduces the signal of power that norm violators' apparent volitional capacity sends, we should find an interaction between volition and the comparison of sanctioned vs. non-sanctioned norm violators, which is why we chose the latter as reference group. This regression revealed a significant effect of volition, $B = 0.50$, $SE = 0.14$, $t(126) = 3.50$, $p = .001$, 95% CI [0.218, 0.786], an interaction between volition and norm abidance (vs. non-sanctioned norm violation), $B = -0.43$, $SE = 0.21$, $t(126) = -2.09$, $p = .039$, 95% CI [-0.838, -0.022], and the expected interaction between volition and sanctioned norm violation (vs. non-sanctioned norm violation), $B = -0.41$, $SE = 0.19$, $t(126) = -2.23$, $p = .028$, 95% CI [-0.780, -0.046]. This suggests that the relationship between volition and power was different for non-sanctioned norm violators compared to both norm abiders and sanctioned norm violators. As the simple slopes in Fig 7 illustrate, for non-sanctioned norm violators, greater volition inferences translated into greater inferences of power, $B = 0.50$, $SE = 0.14$, 95% CI [0.218, 0.786], whereas this was not the case for norm abiders, $B = 0.07$, $SE = 0.15$, 95% CI [-0.222, 0.365], or sanctioned norm violators, $B = 0.09$, $SE = 0.12$, 95% CI [-0.144, 0.322]. The positive slope for non-sanctioned norm violators significantly differed from the flatter slopes of both norm abiders, $t(126) = 2.09$, $p = .039$, 95% CI [0.022, 0.838], $d = 0.453$, and sanctioned norm violators, $t(126) = 2.23$, $p = .028$, 95% CI [0.046, 0.780], $d = 0.476$, indicating that sanctions indeed attenuated the signal of power that norm violator's apparent volition sends.

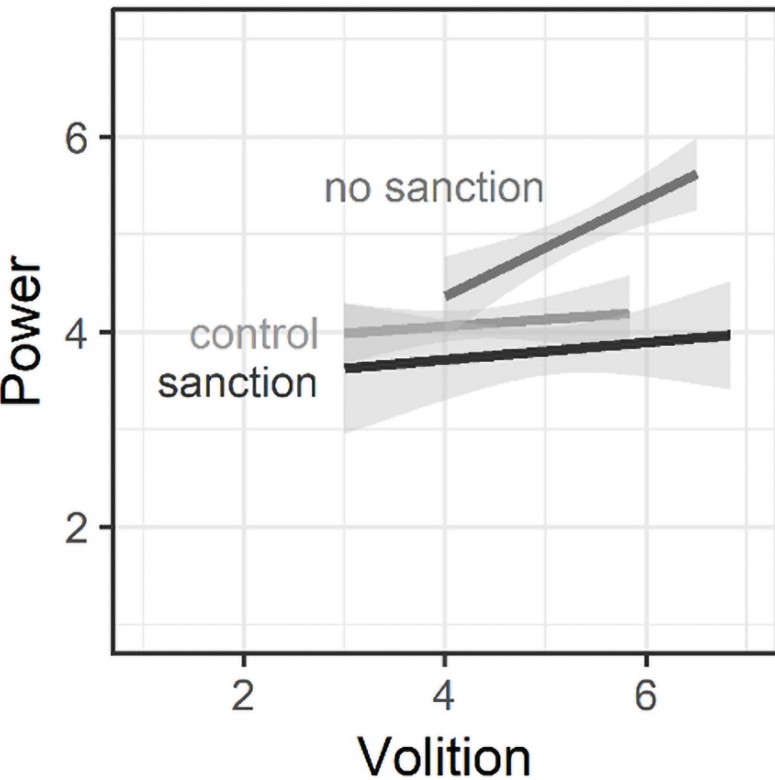

**Fig 7. Simple slopes for the interaction of volition and sanctioning on power perceptions.** The labels in the figure correspond to the following labels in Study 1: no sanction (non-sanctioned norm violator in Study 1), sanction (sanctioned norm violator), and control (non-sanctioned norm abider).

## Discussion

Study 2 replicated the finding that norm violators are seen as acting more according to their own volition, and that greater volition in turn relates to greater inferences of power [5]. As expected, sanctioning reduced the extent to which norm violators were seen as powerful, but it did not significantly affect the extent to which norm violators appeared to act according to their own volition. This suggests that sanctioning specifically targets the power-signaling effect of volition, and the interaction between experimental condition and volition further supported this prediction.

## General discussion

Next to eliciting negative responses in observers, people who violate norms also demonstrate that they can behave as they wish, which makes them appear powerful [5]. This may open the door to a "self-reinforcing loop" (p. 351 [16]) in which norm violators gain power in the eyes of observers, in turn giving norm violators more leeway to keep violating norms and consolidating their ascribed power. The question then arises: How can we prevent people from gaining influence through norm violations? Here we proposed that sanctioning reduces the extent to which norm violator's volition signals power, thereby breaking the norm violation → volition → perceived power chain. In two studies we replicated this chain [5], and in both studies sanctions reduced perceptions of power. In Study 1, in which we prioritized the use of a full-factorial design over ecological validity, sanctioning reduced power perceptions irrespective of

volition. In Study 2, in which we employed a one-factor design to enable creating more realistic scenarios, we found support for the idea that sanctioning specifically targets the extent to which norm violators' apparent volition signals power. Together, the results of both studies suggest that sanctioning can break the self-reinforcing loop to power that norm violations might set off [5, 16].

## Theoretical and practical implications

The current findings have a number of implications. From a theoretical perspective, we demonstrated that sanctioning reduces power perceptions, rather than perceptions of volition. By identifying a boundary condition of the power-signaling effect of volition, we expand previous research on this link [5, 31] and enrich understanding of costly signaling [25, 26]. Our findings suggest that potentially costly behavior (e.g., a norm violation) can only act as a signal of an underlying trait (e.g., power) in the absence of additional cues that provide direct information about that trait (e.g., no sanctions). When translating costly signaling theory from animal to human behavior [25, 40], the possibility that additional information (e.g., a sanction) may drown potentially costly indirect signals (e.g., the demonstration of volitional capacity) needs to be taken into account.

From a practical perspective, our findings suggest that sanctions may be effective in breaking the self-reinforcing loop to power that norm violations may set off [5, 16]. This points to ways in which the ascent of norm violators in social hierarchies can be prevented. For example, employees can create a culture in which blatant interruptions are not tolerated by reprimanding interrupters. Should norm violations persist, more formal sanctions may be called for.

## Limitations and future directions

The current study has a number of limitations. First, although in both studies sanctioning reduced power perceptions, the results are mixed concerning the underlying mechanism. Whereas in Study 1 sanctioning reduced power perceptions irrespective of volition, Study 2 yielded support for the idea that sanctioning specifically targets the extent to which norm violators' apparent volition signals power. One explanation for this discrepancy may lie in the different control conditions we employed. In Study 1, norm abiders were—like norm violators—not able to show a valid ticket, and some norm abiders were also sanctioned. Although this design is adequate to test predictions in a full-factorial model allowing different comparisons between conditions, it also made interpretation of the results difficult. We solved this dilemma by running a second study that was more realistic and unequivocal as norm abiders now bought and showed a valid ticket to the controller. Future replication efforts should therefore focus on Study 2 to gain further confidence in the robustness of our findings. Also, although previous research [5, 14] confirmed the mediating role of volition in the link between norm violation and perceived power, future research could experimentally manipulate volition as to substantiate a causal relation between volition and perceived power.

A second limitation is our reliance on scenarios. This approach affords experimental control and allowed us to make clear to our participants whether or not norms were violated (by informing participants whether a ticket was bought). Although previous research [5, 8, 23, 24] has shown that results obtained in scenario studies were very consistent with results obtained in more realistic settings, future studies could investigate and extend the current findings using more ecologically valid procedures. In addition, strong evidence for the effect of norm

violation on power perceptions would be if bystanders would submit to the supposed power of norm violators, for example, by following their instructions. Future research could focus on measuring the behaviors of bystanders reflecting their submission to norm violators' power.

A further complication and next step for future research is that real-life interactions may not terminate after a sanction, but instead the norm violator may object to, or even retaliate against, the punisher. Indeed, previous research already pointed out that enacting sanctions may only be possible for dominant individuals [41], and characteristics of the punisher therefore also need to be taken into account. Also, future studies could investigate observer responses in situations where the norm violator is a member of an ingroup or outgroup or where norm violators continue their behavior after being sanctioned.

Third, we considered the norm violation in this study as a violation of a descriptive and injunctive legal norm. We assumed that buying a train ticket is a well-known legal norm enacted and endorsed as appropriate by most study participants. Although we did not test this assumption, the results of the manipulation checks in both studies showed that participants perceived the behavior of the norm violator to be violating of norms. Train passengers who do not buy a train ticket transgress a legal norm and run the risk of being formally penalized by means of a fine. Note that laws, as opposed to social norms, are not negotiated through social interaction, which means that people's responses to violating the legal norm to buy a train ticket may be relatively similar across social contexts [42]. Prior research has shown that legal norm violations such as financial fraud [5] or illegal parking [23] elicit similar responses from observers as non-legal norm violations such as arriving late to a meeting [8] or putting one's feet on another's table [5]. The recurring pattern across these and various other behaviors is that norm violators are perceived by others as powerful. Future research on norm violation could pay more attention to the actual endorsement and enactment of specific norms among study participants. Additionally, future research could examine situations where the violation of an injunctive norm does not constitute a violation of a descriptive norm and vice versa [43, 44] to understand how participants differentially respond to violations of such more complicated normative influences.

In addition, not all norm violations are created equal [4]. Free-riding on the train is costly to society, and therefore sanctioning may be in order. However, some norms are outright harmful [45]. Going against such harmful norms may underline norm violators' apparent conviction of what is right and wrong. When norms are violated for deontological reasons, sanctioning might not reduce inferences of power. On the contrary, sanctions might elevate norm violators to the status of a martyr as they suffer for a cause [46], thereby allowing them to amass even more influence.

Finally, our studies comprised a majority of female participants from different countries (Germany and the Netherlands). Although this gender composition is not representative for the population, we do not assume gender differences in individual responses to the violation of a legal norm such as buying a train ticket. Moreover, participants were randomly assigned to conditions and our findings corroborate those of previous research. We found that the German participants in Study 1 perceived non-sanctioned norm abiders to have violated norms to a greater extent than sanctioned norm abiders. This unexpected finding might stem from a culturally defined norm that a monetary fine should always be imposed when travelers cannot show a ticket. Indeed, cultures vary in norm strength and tolerance of deviant behavior [47] and may therefore differ in responses to (missing) sanctions. Also, the current studies were conducted in individualistic (as opposed to collectivistic) cultures where there is a positive link between norm violation and power perceptions [8]. Therefore, future research could address possible cultural differences in responses to the sanctioning of norm violations [48].

## Conclusion

Our results indicate that sanctioning can prevent norm violators from gaining power in the eyes of observers. Sanctions may therefore be effective in breaking the self-reinforcing loop to power that norm violations can set off [5, 16].

## Supporting information

**S1 File. Supplemental material.**
(PDF)

## Author Contributions

**Conceptualization:** Florian Wanders, Astrid C. Homan, Annelies E. M. van Vianen, Rima-Maria Rahal, Gerben A. van Kleef.

**Data curation:** Florian Wanders.

**Formal analysis:** Florian Wanders.

**Investigation:** Florian Wanders.

**Methodology:** Florian Wanders, Astrid C. Homan, Rima-Maria Rahal, Gerben A. van Kleef.

**Project administration:** Annelies E. M. van Vianen.

**Resources:** Florian Wanders, Rima-Maria Rahal.

**Supervision:** Astrid C. Homan, Annelies E. M. van Vianen, Gerben A. van Kleef.

**Visualization:** Florian Wanders.

**Writing – original draft:** Florian Wanders.

**Writing – review & editing:** Astrid C. Homan, Annelies E. M. van Vianen, Rima-Maria Rahal, Gerben A. van Kleef.

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
