## [Decision Letter · Decision Letter 0]

18 Feb 2021

PONE-D-20-25007

How norm violators rise and fall in the eyes of others: The role of sanctions

PLOS ONE

Dear Dr. van Vianen,

Thank you for submitting your manuscript to PLOS ONE. After careful consideration, we
feel that it has merit but does not fully meet PLOS ONE’s publication criteria as it
currently stands. Therefore, we invite you to submit a revised version of the
manuscript that addresses the points raised during the review process.

In the revised version of the paper, please address the reviewers' comments listed
below. Additionally, please better explain the meaning of the variables you have
used to validate the two studies.

Please submit your revised manuscript by Apr 04 2021 11:59PM. If you will need more
time than this to complete your revisions, please reply to this message or contact
the journal office at plosone@plos.org. When
you're ready to submit your revision, log on to https://www.editorialmanager.com/pone/ and select the 'Submissions
Needing Revision' folder to locate your manuscript file.

If you would like to make changes to your financial disclosure, please include your
updated statement in your cover letter. Guidelines for resubmitting your figure
files are available below the reviewer comments at the end of this letter.

We look forward to receiving your revised manuscript.

Kind regards,

Camelia Delcea

Academic Editor

PLOS ONE

Journal Requirements:

2. During our internal checks, the in-house editorial staff noted that you conducted
research or obtained samples in another country (for study 1). Please check the
relevant national regulations and laws applying to foreign researchers and state
whether you obtained the required permits and approvals.

Please address this in your ethics statement in both the manuscript and submission
information.

3. Thank you for your ethics statement:

'Institutional review board: Ethical review board, Faculty of Social and Behavioral
Sciences, University of Amsterdam

Approval numbers: 2014-WOP-3498 and 2017-COP-8050

Consent obtained written online anonymously'

a. Please amend your current ethics statement to confirm that your named
institutional review board or ethics committee specifically approved this study.

Reviewers' comments:

Reviewer's Responses to Questions

**Comments to the Author**

1. Is the manuscript technically sound, and do the data support the conclusions?

Reviewer #1: Yes

Reviewer #2: No

Reviewer #3: Partly

Reviewer #4: No

2. Has the statistical analysis been performed
appropriately and rigorously? 

Reviewer #1: Yes

Reviewer #2: Yes

Reviewer #3: Yes

Reviewer #4: Yes

3. Have the authors made all data underlying the
findings in their manuscript fully available?

Reviewer #1: Yes

Reviewer #2: No

Reviewer #3: Yes

Reviewer #4: Yes

4. Is the manuscript presented in an intelligible
fashion and written in standard English?

Reviewer #1: Yes

Reviewer #2: Yes

Reviewer #3: Yes

Reviewer #4: Yes

5. Review Comments to the Author

Reviewer #1: The paper tackles an interesting and important topic, it is well written
and all analyses are described clearly, including limitations and indicating which
aspects where exploratory or previously hypothesized.

I thus have only a few minor comments:

- the authors remain a bit silent about which online participant pools they use. Is
this something comparable to MTurk, or are these student participants?

- Some of the arguments for the difference in effects between the first study and
previous studies are based on arguments about the German Railway norms, however, the
second study then is (again, as previous studies, I guess) done in the Netherlands.
It would be interesting to understand whether this - as opposed to the design
differences - plays a role for the effects, whether norms and norm abidance are
perceived differently in the two systems.

- the authors themselves discuss that one problem with the results is the
hypothetical character. There are by now several studies from the team of
Marie-Claire Villeval (Lyon) which use real settings to study similar questions. I
am wondering whether the authors are aware of this research and whether they might
consider doing more real world studies in the future.

- in their motivational examples, the authors always use firm contexts (interrupting
colleagues etc.) - why do they then choose railway examples for the study?

- my main issue is actually with the signaling idea. If norm violation signals power
through being a potentially costly volitional behavior, sanctioning should not
necessarily reduce perceived power. The study actually cannot scrutinize this link,
as it is a one-shot behavior that is being described. If there were no sanctions, it
wouldn´t be costly signaling. Thus, the described mechanism could only work if norm
violators keep violating even though there is a chance of being sanctioned - which
implies, that in some cases they will be fined, in others not. The design is as it
is, but I would like to see a more thorough discussion of this.

Reviewer #2: The study deals with an interesting and important topic related to
social norms. The methods sound appropriate to test the hypotheses.

I now focus on issues that would help to improve this manuscript:

- One major issue with this paper is the need to explicate social norms. The
literature has been well-documented with norms being conceptualized as injunctive
norms and descriptive norms. The association between norms and social sanction has
been extensively discussed in the work of Cialdini et al. (1990), Fishbein and Ajzen
(2011), and Lapinski and Rimal (2005). Injunctive norms refer to what ought to be
done while descriptive norms pertaining to the prevalence of a behavior. Thus, this
study seems to intend to deal with injunctive norms rather than descriptive norms.
Further, social norms and law are distinct concept (see Rimal & Lapinski, 2015).
This study does not seem to distinguish law violation from norm violation. I would
think this paper focuses on law and legal sanction, rather a norm-based
approach.

- Similarly, the explication of the power concept is limited. Authors define power as
the perceived potential to influence others, which is not real power (individuals
might not actually hold that power, but only are perceived by others). The lack of
explication makes the conceptualization and operationalization of this variable
sound less convincing. When we read/see a person not buying a bus ticket, there is
little ground to argue that others would think the violator has a great deal of
power. The authors use an example of people violating the talking norms in meetings
to illustrate their point, but these two contexts are fundamentally different: Some
people can talk freely in meetings because they either have real power or the
behavior could actually be part of the organizational norms (the meeting norm is
that you can interrupt others' talk if you do have something important to say). I
would not think a traveler who did not conform to the law as having "a great deal of
power," not mentioning that they told lies to authorities. I would think that
someone escaping a law sanction as a lucky individual and that should be inferred as
the person having power, unless he/she has further actions (ex: making a phone call
to powerful others, which is a form of reference power). Such definition and
operationalization as written in the paper, therefore, do not sound convincing to
me.

- I see that it is quite controversial to argue that freedom to do something would
always lead to inferences of power possession. It might only signal power as the
authors suggest under some certain conditions (there should be boundary conditions).
I would think that people can think of someone who acts as she/he wants, which
deviates from social approval, as having less power. This rival theory can be
illustrated, for example, by historical accounts related to social movements in
which less powerful individuals in a society (both real and perceived) violate a
political norm/law to gain power. We may also see drivers overspeed and think of
them as traffic violators who would likely confront more powerful others
(policemen). This social comparison will likely lead to perceptions of the violators
as being less powerful, or even having no power and thus defying law to satisfy
their desired power. In the same vein, a person did not buy a ticket might mean
he/she has no other choice (lack of freedom) and thus violates the law (no power).
This line of reasoning shows that the theorization of the model in this paper seems
problematic because the authors left too many rival theories unaddressed.

- When theorization is not sound, having supporting data does not help much. The
three key variables likely often have some sorts of correlations. A statistical
model can be statistically significant without any theoretical background. Plus, the
idea that someone has power could have more freedom to do things, even violating a
law is not new in the literature. Also, the idea of sanctioning someone reducing
his/her power offers no novel theoretical implication (someone goes to jail of
course will normally have much less power than before). I do not see how such
theorizations add to the literature.

That says, I commend the authors on engaging in a project with a rigorously
methodological design. I sincerely appreciate the author(s)’ work, and I wish them
the best of luck with this project.

Reviewer #3: This manuscript reports two experiments designed to test the hypothesis
that norm violators will appear less powerful when they are punished than when they
are not. The experiments build on earlier research showing that norm violators are
perceived as more powerful than norm abiders; they introduce sanctions as a
moderator of this effect.

The manuscript has a number of strengths: The research is methodologically sound; the
analyses are appropriate; the write-up is clear and complete. At the same time, the
research makes a very modest contribution to the literature, even more modest than
the write-up suggests. It mainly shows that an effect previously demonstrated by
these investigators has limited scope. It is good to know that, of course, but it
does not represent the level of contribution typically found in PLOS ONE
articles.

Let me describe briefly how I would interpret the results of this research, as my
interpretation is somewhat different from how the authors frame the results. These
results demonstrate that in a situation in which there are rules for how to behave,
people are sensitive to where the power lies: with the rules or with the
individuals. (I’m using rules here, rather than norms, because the research scenario
conflates the two, but the same analysis holds for norms.) To the extent that people
follow the rules and violations of the rules are enforced, power lies with the
rules; to the extent that people violate the rules and get away with it, power lies
with the individuals. Volition, on the other hand, depends on whether people try to
follow the rules. Study 1 shows that neither rule-abiders nor rule-violators have
much power if the rules are enforced, but if the rules are not enforced, even people
who accidentally violate them (by not having their ticket to present to the
conductor) have power. Study 2 shows that rules are powerful when people abide by
them and when they are enforced; the relationship between individual volition and
power is strongest when rules are weak. This summary captures all of the findings of
this research and is entirely consistent with current views of how social rules and
norms work. They clarify that the earlier finding of greater power attributed to
norm violators holds only when norms are weak, but that simply serves to limit the
scope and importance of the earlier finding. It does not challenge or extend current
understandings of the way norms work.

I will leave to the editor the decision of whether this manuscript makes enough of a
contribution to warrant publication in PLOS ONE. Regardless of where it is
published, I think some revision is in order to simplify and clarify the
presentation and interpretation of the results.

Reviewer #4: Referee report: PONE-D-20-25007

Summary of the paper

The authors (1) replicate previous research that third-party observers believe that
norm violators have a greater volition and power than norm abiders, and (2) extend
that research to understand whether sanctions can be used to reduce perceptions of
power associated with norm violation. The authors conduct two studies: (1) with a
German online sample and a 2X2 design (Abiding norm, violating norm)X(Sanctions, No
Sanctions) and (2) with a Dutch online sample with 1X3 design (Abiding Norm),
(Violating Norm X Sanctions), (Violating Norm X No Sanctions).

The authors use a vignette about a passenger buying or not buying a ticket on a
train, and a controller either sanctioning (or not) the passenger who fails to show
a ticket. They measure survey respondents’ perceptions of the passenger’s volition
and power using survey questions. The authors claim that the main mechanism of how
norm violation affects power is through volition i.e., a passenger who violates a
norm is considered to act on their own volition, and this belief about volition
leads to increased perceptions about the power they possess. The authors find that
sanctions reduce the perceptions of power of the passenger irrespective of whether
they are norm abiding or not, and irrespective of their volition (Result of Study
1). They find weak evidence for the mechanism that norm violation affects power
through volition.

The paper is well-written, and the data collection and analysis are well-done.

Major critique

1. The paper clearly shows that introduction of sanctions reduces power associated
with both norm-abiding and norm-violating individuals. However, the mechanism that
norm violation leads to increased volition that further leads to increased power is
not supported by evidence. The authors cannot claim that the mechanism is true
unless they vary volition exogenously and find that perceptions of power are
affected by that variation.

2. The results from Study 1 suggest that introduction of sanctions reduce perceptions
of passenger’s power irrespective of his/her volition and his/her norm
abidance/violation. In study 2, the authors find a different result because that
they do not have a treatment with sanctions for norm-abiding behavior in this Study
and thus do not have much variation in volition. I don’t think we can conclude from
Study 2 that the claimed mechanism (Norm ViolationIncreased
volitionIncreased power) is true.

3. Moreover, volition and power are correlated. However, there is no evidence that it
is higher volition that leads to greater power. It could be the other way round
where higher power leads to having greater volition.

4. It is not clear what “power” means in the context of a passenger who either buys
or does not buy a ticket on a train. How does not buying a ticket make one more
influential? A better way to measure power in this situation would be to see if the
third-party observer is more likely to follow instructions from someone who violated
the norm versus who obeyed the norm.

5. The payment to participants is small and probabilistic. For example, the
participants had a 20% chance of winning a 10Euro voucher in Study 2. It is unclear
how seriously the participants took the survey with these small incentives.

6. The authors use a vignette about the norm of buying a ticket or not on a train.
The authors may want to discuss how this specific situation can be generalized to
other situations.

Minor critique

1. When you say norms, can you clarify if they are descriptive or prescriptive
norms?

2. Both the sanction and no sanction conditions in the paper technically have
sanctions, in one case they are enforced and in another they are not. The authors
can clarify that by changing their terminology.

3. 75% of the sample is women and is not representative of the German population in
Study 1. The authors may want to discuss how the gender composition of their sample
may affect the result.

4. Since Study 1 and 2 are conducted with different populations (German vs Dutch
online samples), the others should comment on how comparable these studies are. Are
there differences in norms of ticket buying in these two populations?

6. PLOS authors have the option to publish the peer
review history of their article (what does this mean?). If published, this will
include your full peer review and any attached files.

If you choose “no”, your identity will remain anonymous but your review may still be
made public.

**Do you want your identity to be public for this peer review?** For
information about this choice, including consent withdrawal, please see our
Privacy Policy.

Reviewer #1: No

Reviewer #2: No

Reviewer #3: No

Reviewer #4: No

---

## [Author Response · Author response to Decision Letter 0]

28 Mar 2021

Responses to Editor comments

Response: We have carefully checked that our manuscript meets PLOS ONE's style
requirements.

2. During our internal checks, the in-house editorial staff noted that you conducted
research or obtained samples in another country (for study 1). Please check the
relevant national regulations and laws applying to foreign researchers and state
whether you obtained the required permits and approvals. Please address this in your
ethics statement in both the manuscript and submission information.

Response: The study was approved by the Ethical review board of the Faculty of Social
and Behavioral Sciences at the University of Amsterdam, but data was collected
online from a convenience sample of participants in Germany. The Code of Conduct of
the German Psychological Society stipulates ethical considerations (https://www.dgps.de/index.php?id=85) for research
with human participants, which do not reference special permits required for
international researchers. Further, ethical considerations in research in Germany
are subject to the same codified ethical guidelines as in the Netherlands, namely
the Helsinki Declaration and European data protection regulations. Therefore, no
additional permits were required to conduct this research. We have added this
information in the ethics statement in both the manuscript (Methods section Study 1)
and submission.

3. Thank you for your ethics statement:

'Institutional review board: Ethical review board, Faculty of Social and Behavioral
Sciences, University of Amsterdam, Approval numbers: 2014-WOP-3498 and
2017-COP-8050

Consent obtained written online anonymously'

a. Please amend your current ethics statement to confirm that your named
institutional review board or ethics committee specifically approved this study.

Response: We have amended the ethics statements in the Methods sections of Study 1
(p. 6) and Study 2 (p. 14), respectively and we have added the same text to the
“Ethics Statement” field of the submission form.

Responses to Reviewer #1: 

The paper tackles an interesting and important topic, it is well written and all
analyses are described clearly, including limitations and indicating which aspects
where exploratory or previously hypothesized.

I thus have only a few minor comments:

Response: Thank you for the overall positive evaluation of our work and your
constructive comments. 

1. the authors remain a bit silent about which online participant pools they use. Is
this something comparable to MTurk, or are these student participants?

Response: The online participant pool (Mage=23.78) was recruited from passers-by at a
German university campus as well as through social media. We have added this
information on page 6.

2. Some of the arguments for the difference in effects between the first study and
previous studies are based on arguments about the German Railway norms, however, the
second study then is (again, as previous studies, I guess) done in the Netherlands.
It would be interesting to understand whether this - as opposed to the design
differences - plays a role for the effects, whether norms and norm abidance are
perceived differently in the two systems.

Response: There are cultural differences as well as similarities between the two
countries on dimensions that are relevant to norm violation. Germany scores higher
than the Netherlands on cultural tightness, which relates to the importance that is
attached to rules and the severity of punishment for violations (Gelfand et al.,
2011). In Study 1 (German sample) we found that non-sanctioned norm abiders were
perceived to have violated norms to a greater extent than sanctioned norm abiders,
although the effect size was small. We could speculate that participants were aware
of the railway operators’ right to fine the traveler and – due to their tightness -
were indignant that the traveler got away without this fine.

Germany and the Netherlands are comparable in terms of individualism, the degree to
which uniqueness, personal achievement, and self-expression are valued (as opposed
to group harmony and collective outcomes). A cross-cultural comparison of responses
to norm violations (including data from Germany and the Netherlands) revealed that
particularly respondents from individualistic cultures (including Germany and the
Netherlands) perceive norm violators as more powerful than norm abiders (Stamkou et
al., 2019). 

Finally, regarding the specific scenario in this study, we would like to note that
Germany and the Netherlands are European Schengen states, allowing free traveling
among these states, and therefore use similar terms and conditions for train travel.
In the discussion section, we now address the possibility of cultural differences as
a subject for future research. 

3. the authors themselves discuss that one problem with the results is the
hypothetical character. There are by now several studies from the team of
Marie-Claire Villeval (Lyon) which use real settings to study similar questions. I
am wondering whether the authors are aware of this research and whether they might
consider doing more real world studies in the future.

Response: The work by Villeval and her colleagues (e.g., Dai et al., 2018) is very
interesting and employs creative methods. Importantly, this work is mainly focused
on the intrapersonal drivers of norm breaking behaviors (e.g., what determines
cheating behavior in individuals) and not on interpersonal dynamics (i.e., how do
others respond to observing individuals who violate the norms), which is the
approach we take in our work.

Responses to norm violations can indeed be studied in various ways, which entail
different trade-offs between ecological validity and experimental control. We have
used a variety of approaches in our work, including scenarios, pictures, video
clips, recalled situations, and live interactions with trained actors to investigate
responses to norm violations (Stamkou et al., 2016, 2018, 2019; Van Kleef et al.,
2011, 2012). Scenarios, pictures, and video clips afford greater experimental
control, whereas recalled situations and live interactions afford greater ecological
validity. Which method is most suitable in a given study depends on the nature of
the research question in combination with the possibilities and constraints of the
different methodological approaches. For the current project, we prioritized
experimental control to enable causal conclusions about the effects of sanctioning
on responses to norm violators. Although we acknowledge the limitations of the
scenario approach, we have found in our previous work that results obtained in
scenario studies were very consistent with results obtained in richer yet less
controlled settings. We are therefore confident in the validity of the current
findings. Nonetheless, we see value in validating and extending the current findings
using more ecologically valid procedures, and in the revised paper we explicitly
call for future research using such procedures (see p. 22).

References (not mentioned in the paper):

Dai, Z., Galeotti, F., & Villeval, M. C. (2018). Cheating in the lab predicts
fraud in the field: An experiment in public transportation. Management Science,
64(3), 1081-1100.

Stamkou, E., Van Kleef, G. A., & Homan, A. C. (2018). The art of influence: When
and why deviant artists gain impact. Journal of Personality and Social Psychology,
115, 276-303.

4. in their motivational examples, the authors always use firm contexts (interrupting
colleagues etc.) - why do they then choose railway examples for the study?

Response: We have used the railway scenario because all study participants can easily
imagine this setting and most likely have experience with the described situation.
In our previous work (see references comment 3), we have used organizational,
educational, artistic, and personal settings to study responses to norm violators
and found that effects were consistent across settings.

5. my main issue is actually with the signaling idea. If norm violation signals power
through being a potentially costly volitional behavior, sanctioning should not
necessarily reduce perceived power. The study actually cannot scrutinize this link,
as it is a one-shot behavior that is being described. If there were no sanctions, it
wouldn´t be costly signaling. Thus, the described mechanism could only work if norm
violators keep violating even though there is a chance of being sanctioned - which
implies, that in some cases they will be fined, in others not. The design is as it
is, but I would like to see a more thorough discussion of this.

Response: There is considerable evidence that individuals who violate norms are
perceived by others as powerful (for a recent review, see Stamkou, Homan, & Van
Kleef, 2020). The theoretical rationale underlying this prediction is that people
who violate norms signal that they experience the leeway to act as they please
despite normative constraints (Van Kleef et al., 2011). This is a freedom that
typically comes with higher rank (Galinsky, Magee, Gruenfeld, Whitson, &
Liljenquist, 2008). Accordingly, research has shown that people who violate norms
are perceived as having high power (e.g., Van Kleef et al., 2011), status (e.g.,
Bellezza et al., 2014), and influence (e.g., Stamkou et al., 2018). Regarding the
used scenario, individuals who do not buy a train ticket run a (high) risk of being
sanctioned. Therefore, by not buying a ticket they signal to be oblivious to
normative constraints and can behave as they wish (volition typically reserved for
the powerful), which elicit power perceptions in observers. Our study shows that
sanctions attenuated the signal of power that norm violator’s apparent volition
sends. Observers likely conclude that the sanctioned norm violator does not have the
power he seemed to have. In future studies, it would indeed be interesting to
investigate how observers will respond to norm violators who keep violating even
after having been sanctioned. We added this suggestion to the general discussion
section.

References (not mentioned in the paper):

Stamkou, E., Homan, A. C., & Van Kleef, G. A. (2020). Climbing the ladder or
falling from grace? A threat-opportunity framework of the effects of norm violations
on social rank. Current Opinion in Psychology, 33, 74-79. 

Galinsky, A. D., Magee, J. C., Gruenfeld, D. H, Whitson, J. A., & Liljenquist, K.
A. (2008). Power reduces the press of the situation: Implications for creativity,
conformity, and dissonance. Journal of Personality and Social Psychology, 95,
1450–1466.

Responses to Reviewer #2: 

The study deals with an interesting and important topic related to social norms. The
methods sound appropriate to test the hypotheses.

I now focus on issues that would help to improve this manuscript:

Response: Thank you for this positive comment to our work and your suggestions for
improvement. 

1. One major issue with this paper is the need to explicate social norms. The
literature has been well-documented with norms being conceptualized as injunctive
norms and descriptive norms. The association between norms and social sanction has
been extensively discussed in the work of Cialdini et al. (1990), Fishbein and Ajzen
(2011), and Lapinski and Rimal (2005). Injunctive norms refer to what ought to be
done while descriptive norms pertaining to the prevalence of a behavior. Thus, this
study seems to intend to deal with injunctive norms rather than descriptive norms.
Further, social norms and law are distinct concept (see Rimal & Lapinski, 2015).
This study does not seem to distinguish law violation from norm violation. I would
think this paper focuses on law and legal sanction, rather a norm-based
approach.

Response: Although there is a clear conceptual distinction between descriptive and
injunctive norms, many common norm violations fall in both categories (Van Kleef,
Gelfand, & Jetten, 2019). This is because behaviors that are endorsed as
appropriate by the majority of the members of a group (injunctive norms) also tend
to be enacted by the majority of the members of a group (descriptive norms).
Accordingly, almost all prior research on responses to norm violators has examined
behaviors that would be considered violations of both descriptive and injunctive
norms. For instance, studies examined responses to individuals who would come in
late for a work meeting (Stamkou et al., 2019), put their feet on someone else’s
table (Van Kleef et al., 2011), take someone else’s coffee (Van Kleef et al., 2012),
park their bike in an illegal spot (Stamkou et al., 2016), or dress improperly for a
(professional) occasion (Bellezza, Gino, & Keinan, 2014; Oostrom, Ronay, &
Van Kleef, 2021). These are all behaviors that simultaneously infringe on injunctive
norms (most people disapprove of these behaviors) and descriptive norms (most people
do not exhibit these behaviors). The same is true for the norm violation examined in
the current work: Most people believe it is appropriate to buy a train ticket
(injunctive norm) and most people indeed do so (descriptive norm). The current
operationalization thus reflects the natural conflation of descriptive and
injunctive norms in real life. That said, it would be interesting to investigate in
future research whether the moderating effect of sanctioning also applies to “pure”
violations of descriptive versus injunctive norms. We now refer to descriptive and
injunctive norms in the introduction section and we have added this suggestion for
future research in the general discussion section.

The reviewer is correct in noting that we did not draw an explicit distinction
between social and legal violations. Conceptually, we see legal violations as a
subset of the broader category of norm violations. That is, people may violate norms
in ways that are or are not punishable, depending on the nature of the infringement
and the broader context (e.g., a national law system). In our previous work, we have
seen that legal norm violations such as financial fraud (Van Kleef et al., 2011) or
illegal parking (Stamkou et al., 2016) elicit similar responses from observers as
non-legal norm violations such as arriving late to a meeting (Stamkou et al., 2019)
or putting one’s feet on another’s table (Van Kleef et al., 2011): The recurring
pattern across these and various other behaviors is that norm violators are
perceived by others as powerful. In the revised introduction, we have made it
explicit that our focus in the current research was on legal norm violations.
Additionally, in the general discussion section (limitations and future directions)
we note that previous research indicates that legal and non-legal norm violations
elicit similar social responses (p. 23). 

References (not mentioned in the paper):

Oostrom, J. K., Ronay, R., & Van Kleef, G. A. (2021). The signalling effects of
nonconforming dress style in personnel selection contexts: Do applicants’
qualifications matter? European Journal of Work and Organizational Psychology, 30,
70-82.

2. Similarly, the explication of the power concept is limited. Authors define power
as the perceived potential to influence others, which is not real power (individuals
might not actually hold that power, but only are perceived by others). The lack of
explication makes the conceptualization and operationalization of this variable
sound less convincing. When we read/see a person not buying a bus ticket, there is
little ground to argue that others would think the violator has a great deal of
power. The authors use an example of people violating the talking norms in meetings
to illustrate their point, but these two contexts are fundamentally different: Some
people can talk freely in meetings because they either have real power or the
behavior could actually be part of the organizational norms (the meeting norm is
that you can interrupt others' talk if you do have something important to say). I
would not think a traveler who did not conform to the law as having "a great deal of
power," not mentioning that they told lies to authorities. I would think that
someone escaping a law sanction as a lucky individual and that should be inferred as
the person having power, unless he/she has further actions (ex: making a phone call
to powerful others, which is a form of reference power). Such definition and
operationalization as written in the paper, therefore, do not sound convincing to
me.

Response: There is considerable evidence that individuals who violate norms are
perceived by others as powerful (for a recent review, see Stamkou, Homan, & Van
Kleef, 2020). The theoretical rationale underlying this prediction is that people
who violate norms signal that they experience the leeway to act as they please
despite normative constraints (Van Kleef et al., 2011). This is a freedom that
typically comes with higher rank (Galinsky, Magee, Gruenfeld, Whitson, &
Liljenquist, 2008). Accordingly, research has shown that people who violate norms
are perceived as having high power (e.g., Van Kleef et al., 2011), status (e.g.,
Bellezza et al., 2014), and influence (e.g., Stamkou et al., 2018). Furthermore,
there is evidence that these perceptions can, under particular circumstances, fuel
actual granting of power, for instance via the conferral of control over outcomes,
voting, and leadership endorsement (Stamkou et al., 2016; Van Kleef et al., 2012).
Our two studies have shown that even individuals who did not buy a train ticket were
perceived as more powerful. To acknowledge that our research speaks to perceived
power, we have made this explicit throughout the paper (e.g., p. 4, pp. 9-13, pp.
15-17, p. 20, p. 22).

References (not mentioned in the paper):

Stamkou, E., Homan, A. C., & Van Kleef, G. A. (2020). Climbing the ladder or
falling from grace? A threat-opportunity framework of the effects of norm violations
on social rank. Current Opinion in Psychology, 33, 74-79.

Galinsky, A. D., Magee, J. C., Gruenfeld, D. H, Whitson, J. A., & Liljenquist, K.
A. (2008). Power reduces the press of the situation: Implications for creativity,
conformity, and dissonance. Journal of Personality and Social Psychology, 95,
1450–1466.

3. I see that it is quite controversial to argue that freedom to do something would
always lead to inferences of power possession. It might only signal power as the
authors suggest under some certain conditions (there should be boundary conditions).
I would think that people can think of someone who acts as she/he wants, which
deviates from social approval, as having less power. This rival theory can be
illustrated, for example, by historical accounts related to social movements in
which less powerful individuals in a society (both real and perceived) violate a
political norm/law to gain power. We may also see drivers overspeed and think of
them as traffic violators who would likely confront more powerful others
(policemen). This social comparison will likely lead to perceptions of the violators
as being less powerful, or even having no power and thus defying law to satisfy
their desired power. In the same vein, a person did not buy a ticket might mean
he/she has no other choice (lack of freedom) and thus violates the law (no power).
This line of reasoning shows that the theorization of the model in this paper seems
problematic because the authors left too many rival theories unaddressed.

Response: Please, see our response to your comment 2 and the recent review of Stamkou
et al., 2020. In our paper we recognize that potentially costly behavior (e.g., a
norm violation) can only act as a signal of an underlying trait (e.g., power) in the
absence of additional cues that provide direct information about that trait (see p.
4). Hence, norm violation may not signal power to the same degree when additional
information is available. Yet, if additional information is lacking (which seems
also the case in your example of the speedy driver), bystanders tend to ascribe
power to the norm violator as has been repeatedly demonstrated in prior research and
in the current study. We stress this in the discussion section on pp. 21-22. To our
best knowledge, our study is one of the few examining a boundary condition, namely
sanctioning. Future research could investigate other boundary conditions, such as
the background of the norm violator (which can act as an additional information
cue). For instance, if people are aware of the power of the person violating the
norm – a situation that you raise in your comment above – this should act as a
moderator of the effect. Distantly speaking to your comment, we have some
unpublished research that suggests that outgroup members breaking the norm are not
seen as more powerful, whereas ingroup members who break the norm are seen as more
powerful. We have added this suggestion to the discussion section (p. 22).

4. When theorization is not sound, having supporting data does not help much. The
three key variables likely often have some sorts of correlations. A statistical
model can be statistically significant without any theoretical background. Plus, the
idea that someone has power could have more freedom to do things, even violating a
law is not new in the literature. Also, the idea of sanctioning someone reducing
his/her power offers no novel theoretical implication (someone goes to jail of
course will normally have much less power than before). I do not see how such
theorizations add to the literature.

Response: The key contribution of our study is not that sanctioning reduces power per
se, but that sanctioning reduces the effect of norm violation on power perceptions.
Our work is based on costly signaling theory (Bergmüller et al., 2007; Zahavi, 1995)
and prior empirical evidence for the norm violation � volition �
power chain (see Stamkou et al., 2020). A robust finding (in individualistic
societies) is that norm violators are perceived by others as powerful and high
status. This is important, because people often defer to (and are less likely to
speak up to) others whom they perceive as powerful (Anderson & Kilduff, 2009;
Cheng, Tracy, Foulsham, Kingstone, & Henrich 2013). Moreover, norm violators are
sometimes granted power and leadership due to the impression they make on others,
and this increased power in turn makes future violations more likely (Van Kleef,
Wanders, Stamkou, & Homan, 2015). To the degree that this vicious cycle is a
cause for concern, it is important to understand how it can be broken. The current
research provides first evidence that sanctioning can sever the link between norm
violation and power perceptions, thereby disrupting a potentially toxic spiral of
norm violation and power abuse. Future research can build on our work and provide
further evidence for the role of sanctions or other punishing responses (e.g.,
informal (social) punishment) in preventing people from gaining influence through
norm violations. 

References (not mentioned in the paper):

Cheng JT, Tracy JL, Foulsham T, Kingstone A, Henrich J. Two ways to the top: evidence
that dominance and prestige are distinct yet viable avenues to social rank and
influence. J Pers. Soc. Psychol. 2013; 104(1). 103–125. doi: 10.1037/a0030398.

That says, I commend the authors on engaging in a project with a rigorously
methodological design. I sincerely appreciate the author(s)’ work, and I wish them
the best of luck with this project.

Response: We appreciate your challenging and constructive comments.

Responses to Reviewer #3: 

This manuscript reports two experiments designed to test the hypothesis that norm
violators will appear less powerful when they are punished than when they are not.
The experiments build on earlier research showing that norm violators are perceived
as more powerful than norm abiders; they introduce sanctions as a moderator of this
effect.

The manuscript has a number of strengths: The research is methodologically sound; the
analyses are appropriate; the write-up is clear and complete. At the same time, the
research makes a very modest contribution to the literature, even more modest than
the write-up suggests. It mainly shows that an effect previously demonstrated by
these investigators has limited scope. It is good to know that, of course, but it
does not represent the level of contribution typically found in PLOS ONE
articles.

Response: Thank you for your positive but also critical general comment. 

Let me describe briefly how I would interpret the results of this research, as my
interpretation is somewhat different from how the authors frame the results. These
results demonstrate that in a situation in which there are rules for how to behave,
people are sensitive to where the power lies: with the rules or with the
individuals. (I’m using rules here, rather than norms, because the research scenario
conflates the two, but the same analysis holds for norms.) To the extent that people
follow the rules and violations of the rules are enforced, power lies with the
rules; to the extent that people violate the rules and get away with it, power lies
with the individuals. Volition, on the other hand, depends on whether people try to
follow the rules. Study 1 shows that neither rule-abiders nor rule-violators have
much power if the rules are enforced, but if the rules are not enforced, even people
who accidentally violate them (by not having their ticket to present to the
conductor) have power. Study 2 shows that rules are powerful when people abide by
them and when they are enforced; the relationship between individual volition and
power is strongest when rules are weak. This summary captures all of the findings of
this research and is entirely consistent with current views of how social rules and
norms work. They clarify that the earlier finding of greater power attributed to
norm violators holds only when norms are weak, but that simply serves to limit the
scope and importance of the earlier finding. It does not challenge or extend current
understandings of the way norms work.

I will leave to the editor the decision of whether this manuscript makes enough of a
contribution to warrant publication in PLOS ONE. Regardless of where it is
published, I think some revision is in order to simplify and clarify the
presentation and interpretation of the results.

Response: Our research is indeed built on solid theory and prior empirical evidence.
We believe it is valuable to test theory in different context and with different
methods. Also, it is good science to examine the boundary conditions of a theory. In
this study, we do both: replicating prior evidence - but now in an experimental
context where a legal norm is violated - and testing a boundary condition
(sanctioning). Indeed, we show that norm violators gain more power in the eyes of
bystanders when norms are not enforced by sanctions, which we believe is a highly
relevant finding for dealing with norm violating behaviors in society. This finding
may also hold for the violation of non-legal norms where the sanction is not a fine
established by law or official rule but rather depends on the responses of
bystanders. We plan further research into the ‘self-reinforcing loop’, by which norm
violators appear powerful, bystanders submit to and consolidate the power of the
norm violators and thus encourage further norm violation, can be broken (see p. 22).
Based on your comment and the comments of the other reviewers we revised the text of
the paper to clarify our conceptualizations and the interpretation of the results
(see pp. 5, 21-24).

Responses to Reviewer #4:

Summary of the paper

The authors (1) replicate previous research that third-party observers believe that
norm violators have a greater volition and power than norm abiders, and (2) extend
that research to understand whether sanctions can be used to reduce perceptions of
power associated with norm violation. The authors conduct two studies: (1) with a
German online sample and a 2X2 design (Abiding norm, violating norm)X(Sanctions, No
Sanctions) and (2) with a Dutch online sample with 1X3 design (Abiding Norm),
(Violating Norm X Sanctions), (Violating Norm X No Sanctions).

The authors use a vignette about a passenger buying or not buying a ticket on a
train, and a controller either sanctioning (or not) the passenger who fails to show
a ticket. They measure survey respondents’ perceptions of the passenger’s volition
and power using survey questions. The authors claim that the main mechanism of how
norm violation affects power is through volition i.e., a passenger who violates a
norm is considered to act on their own volition, and this belief about volition
leads to increased perceptions about the power they possess. The authors find that
sanctions reduce the perceptions of power of the passenger irrespective of whether
they are norm abiding or not, and irrespective of their volition (Result of Study
1). They find weak evidence for the mechanism that norm violation affects power
through volition.

The paper is well-written, and the data collection and analysis are well-done.

Response: Thank you for your positive assessment of our work and your detailed
comments. 

Major critique

1. The paper clearly shows that introduction of sanctions reduces power associated
with both norm-abiding and norm-violating individuals. However, the mechanism that
norm violation leads to increased volition that further leads to increased power is
not supported by evidence. The authors cannot claim that the mechanism is true
unless they vary volition exogenously and find that perceptions of power are
affected by that variation.

Response: There is ample evidence from previous work that individuals who violate
norms are perceived by others as having high volitional capacity, which in turn
fuels perceptions of power and influence (e.g., Bellezza et al., 2014; Stamkou et
al., 2018; Van Kleef et al., 2011). We replicate this link between volition and
power. However, we agree that we didn’t manipulate volition and thus cannot claim
causal evidence for this link. We address this point in the limitations and future
directions section of the manuscript (see p. 22).

2. The results from Study 1 suggest that introduction of sanctions reduce perceptions
of passenger’s power irrespective of his/her volition and his/her norm
abidance/violation. In study 2, the authors find a different result because that
they do not have a treatment with sanctions for norm-abiding behavior in this Study
and thus do not have much variation in volition. I don’t think we can conclude from
Study 2 that the claimed mechanism (Norm ViolationIncreased
volitionIncreased power) is true.

Response: The issue of sanctions for norm-abiding behavior is something we discussed
at length in the author team. On the one hand, from the point of view of having
orthogonal manipulations and allowing different comparisons between conditions, it
is indeed desirable to include a condition in which norm-abiding behavior is
sanctioned. On the other hand, from the point of view of validity, including such a
condition is not desirable as it undermines the credibility of the scenario and
makes the interpretation of comparisons with that condition less straightforward. We
solved this dilemma by running two studies, one of each type, so that the
disadvantages of one approach are remedied by the advantages of the other approach.
We believe that in conjunction the two studies provide reasonable support for the
presumed theoretical mechanism of volitional capacity. We have made these
considerations more explicit in the revision (introduction Study 2 and limitations
in the general discussion section; p. 13 and 21 respectively).

3. Moreover, volition and power are correlated. However, there is no evidence that it
is higher volition that leads to greater power. It could be the other way round
where higher power leads to having greater volition.

Response: Please, see our response to your comment 1.

4. It is not clear what “power” means in the context of a passenger who either buys
or does not buy a ticket on a train. How does not buying a ticket make one more
influential? A better way to measure power in this situation would be to see if the
third-party observer is more likely to follow instructions from someone who violated
the norm versus who obeyed the norm.

Response: There is considerable evidence that individuals who violate norms are
perceived by others as powerful (for a recent review, see Stamkou, Homan, & Van
Kleef, 2020). The theoretical rationale underlying this prediction is that people
who violate norms signal that they experience the leeway to act as they please
despite normative constraints (Van Kleef et al., 2011). This is a freedom that
typically comes with higher rank (Galinsky, Magee, Gruenfeld, Whitson, &
Liljenquist, 2008). Accordingly, research has shown that people who violate norms
are perceived as having high power (e.g., Van Kleef et al., 2011), status (e.g.,
Bellezza et al., 2014), and influence (e.g., Stamkou et al., 2018). Furthermore,
there is evidence that these perceptions can, under particular circumstances, fuel
actual granting of power, for instance via the conferral of control over outcomes,
voting, and leadership endorsement (Stamkou et al., 2016; Van Kleef et al., 2012).
Replicating previous studies, we also found that norm violation - operationalized as
a passenger who did not buy a ticket on a train - elicited perceptions of power.
Since people tend to submit to powerful others (DeRue & Ashford, 2010;
Epitropaki et al., 2013; Van Kleef et al., 2012) it would indeed be valuable to
investigate - in a similar context as in the current study - if observers are also
more likely to follow instructions from norm violators (p. 22). Thank you for this
interesting suggestion. We have added it to the general discussion section.

5. The payment to participants is small and probabilistic. For example, the
participants had a 20% chance of winning a 10Euro voucher in Study 2. It is unclear
how seriously the participants took the survey with these small incentives.

Response: We are confident that our findings are based on data of participants who
took the survey seriously. First, we excluded participants who did not finish the
questionnaire and therefore did not invest enough effort. Second, we excluded
participants who failed the attention checks. Third, the manipulation checks showed
that the intended differences between conditions in the presence or absence of norm
violation and sanctions were achieved. Finally, although we believe that all
participants who completed the questionnaire were sufficiently motivated,
participants were randomly assigned to conditions and thus we may assume that
participants’ motivation is the same across conditions.

6. The authors use a vignette about the norm of buying a ticket or not on a train.
The authors may want to discuss how this specific situation can be generalized to
other situations.

Response: Previous work has revealed that very different norm violations across a
variety of settings have very similar effects on perceptions of power. For instance,
studies examined responses to individuals who would come in late for a work meeting
(Stamkou et al., 2019), put their feet on someone else’s table (Van Kleef et al.,
2011), take someone else’s coffee (Van Kleef et al., 2012), park their bike in an
illegal spot (Stamkou et al., 2016), or dress improperly for a (professional)
occasion (Bellezza, Gino, & Keinan, 2014; Oostrom, Ronay, & Van Kleef,
2021). These are all behaviors that people typically neither approve nor exhibit.
The same is true for the norm violation examined in the current work: Most people
find it appropriate to buy a train ticket and they behave accordingly. All in all,
we believe that the findings reported in the current paper are likely to generalize
to other types of norm violations and other settings.

References (not mentioned in the paper):

Oostrom, J. K., Ronay, R., & Van Kleef, G. A. (2021). The signalling effects of
nonconforming dress style in personnel selection contexts: Do applicants’
qualifications matter? European Journal of Work and Organizational Psychology, 30,
70-82.

Minor critique

1. When you say norms, can you clarify if they are descriptive or prescriptive
norms?

Response: Although there is a clear conceptual distinction between descriptive and
prescriptive (injunctive) norms, many common norm violations fall in both categories
(Van Kleef, Gelfand, & Jetten, 2019). This is because behaviors that are
endorsed as appropriate by the majority of the members of a group also tend to be
enacted by the majority of the members of a group. Accordingly, almost all prior
research on responses to norm violators has examined behaviors that would be
considered violations of both descriptive and injunctive norms (see Bellezza et al.,
2014; Oostrom et al., 2021; Stamkou et al., 2016; Stamkou et al., 2019; Van Kleef et
al., 2011; Van Kleef et al., 2012). The same is true for the norm violation examined
in the current work: Most people believe it is appropriate to buy a train ticket and
most people indeed do so. The current operationalization thus reflects the natural
conflation of descriptive and injunctive norms in real life. We now refer to
descriptive and injunctive norms in the introduction section of the revised
paper.

2. Both the sanction and no sanction conditions in the paper technically have
sanctions, in one case they are enforced and in another they are not. The authors
can clarify that by changing their terminology.

Response: We agree that not buying a train ticket carries the risk of a formal
penalty that may be enforced or not. The sanctioning in our paper refers to a formal
(legal) rather than an informal (social) punishment. Depending on the experimental
condition in Study 1, a sanction was imposed or not when the violator exhibited
sanctionable behavior, that is, could not show a ticket to the controller (see page
14). In the overview of our study (page 5) we now state that sanctions refer to
formal sanctions. 

3. 75% of the sample is women and is not representative of the German population in
Study 1. The authors may want to discuss how the gender composition of their sample
may affect the result.

Response: We agree that the gender composition in Study 1 is not representative for
the German population. However, we have no specific assumptions about possible
gender differences in individual responses to the violation of legal norms.
Moreover, participants were randomly assigned to conditions and the findings of the
two studies are largely the same and in line with previous research. We now discuss
the gender composition of our studies in the general discussion section (p. 23).

4. Since Study 1 and 2 are conducted with different populations (German vs Dutch
online samples), the others should comment on how comparable these studies are. Are
there differences in norms of ticket buying in these two populations?

Response: There are cultural differences as well as similarities between the two
countries in dimensions that are relevant to norm violation. Germany scores higher
than the Netherlands on cultural tightness, which relates to the importance that is
attached to rules and the severity of punishment for violations (Gelfand et al.,
2011). In Study 1 (German sample) we found that non-sanctioned norm abiders were
perceived to have violated norms to a greater extent than sanctioned norm abiders,
although the effect size was small. We could speculate that participants were aware
of the railway operators’ right to fine the traveler and – due to their tightness -
were indignant that the traveler got away without this fine.

Germany and the Netherlands are comparable in terms of individualism, the degree to
which uniqueness, personal achievement, and self-expression are valued (as opposed
to group harmony and collective outcomes). A cross-cultural comparison of responses
to norm violations (including data from Germany and the Netherlands) revealed that
particularly respondents from individualistic cultures (including Germany and the
Netherlands) perceive norm violators as more powerful than norm abiders (Stamkou et
al., 2019). 

Finally, regarding the specific scenario in this study, we would like to note that
Germany and the Netherlands are European Schengen states, allowing free traveling
among these states, and therefore use similar terms and conditions for train travel.
In the discussion section (p. 24), we now address the possibility of cultural
differences as a subject for future research.

to reviewers.docx
---

## [Decision Letter · Decision Letter 1]

21 May 2021

PONE-D-20-25007R1

How norm violators rise and fall in the eyes of others: The role of sanctions

PLOS ONE

Dear Dr. van Vianen,

Thank you for submitting your manuscript to PLOS ONE. After careful consideration, we
feel that it has merit but does not fully meet PLOS ONE’s publication criteria as it
currently stands. Therefore, we invite you to submit a revised version of the
manuscript that addresses the points raised during the review process.

In the revised version of the paper, please try to clarify the aspects related to
descriptive and injunctive norms, how the perceived power has been measured, the
specificity / limitations of the study. When revising the paper, please consider the
reviewers' comments listed at the bottom of the email.

Please submit your revised manuscript by Jul 05 2021 11:59PM. If you will need more
time than this to complete your revisions, please reply to this message or contact
the journal office at plosone@plos.org. When
you're ready to submit your revision, log on to https://www.editorialmanager.com/pone/ and select the 'Submissions
Needing Revision' folder to locate your manuscript file.

If you would like to make changes to your financial disclosure, please include your
updated statement in your cover letter. Guidelines for resubmitting your figure
files are available below the reviewer comments at the end of this letter.

We look forward to receiving your revised manuscript.

Kind regards,

Camelia Delcea

Academic Editor

PLOS ONE

Reviewers' comments:

Reviewer's Responses to Questions

**Comments to the Author**

1. If the authors have adequately addressed your comments raised in a previous round
of review and you feel that this manuscript is now acceptable for publication, you
may indicate that here to bypass the “Comments to the Author” section, enter your
conflict of interest statement in the “Confidential to Editor” section, and submit
your "Accept" recommendation.

Reviewer #1: All comments have been addressed

Reviewer #2: (No Response)

Reviewer #3: All comments have been addressed

2. Is the manuscript technically sound, and do the data
support the conclusions?

Reviewer #1: Yes

Reviewer #2: Yes

Reviewer #3: (No Response)

3. Has the statistical analysis been performed
appropriately and rigorously? 

Reviewer #1: Yes

Reviewer #2: Yes

Reviewer #3: (No Response)

4. Have the authors made all data underlying the
findings in their manuscript fully available?

Reviewer #1: Yes

Reviewer #2: (No Response)

Reviewer #3: (No Response)

5. Is the manuscript presented in an intelligible
fashion and written in standard English?

Reviewer #1: Yes

Reviewer #2: Yes

Reviewer #3: (No Response)

6. Review Comments to the Author

Reviewer #1: Thank you for considering my comments carefully. I do not fully agree
with your take on the signaling part, but your arguments are solid, and it is rather
an empirical question whether your take is right, I think. As you refer it to
further research, I think that´s sufficient.

Reviewer #2: I appreciate the authors’ efforts to address reviewers’ comments. I have
these questions for the authors to clarify:

1. Descriptive and injunctive norms were not clearly defined in the revised
manuscript. These norms are individually perceived. Social norms can be examined at
the collective level, which is different from social norm existing at the
individual/perceived level. At the collective level, both types of norms can
converge, but not necessarily so at the perceived level. People are not always
cognizant of the prevailing descriptive or injunctive norms in certain contexts
(please see Tankard & Paluck, 2016). The social norm approach, therefore,
suggests that misperception of social norms is a problematic issue for
norm-violating behaviors (please see Berkowitz, 2005).

The authors wrote that injunctive norms and descriptive norms almost always work in
the same directions. They wrote that behaviors that are endorsed as appropriate by
the majority of the members of a group (injunctive norms) also tend to be enacted by
the majority of the members of a group (descriptive norms). However, there are many
situations where these two types of normative influences do not overlap, such as
when people approve of, but do not practice, particular behaviors (Cialdini et al.,
1990). Descriptive norms and injunctive norms can also be antagonistic, and they may
provide us with conflicting information about normative behaviors in a given context
(Lapinski & Rimal, 2005). For example, consider these norm-violating behaviors:
drinking, smoking, speeding, etc. (please see, for example, Chung & Rimal, 2016;
Hue et al., 2015).

The authors responded that “almost all prior research on responses to norm violators
has examined behaviors that would be considered violations of both descriptive and
injunctive norms.” Perhaps, this manuscript needs to speak for itself as to why
these two types of norms are almost all considered in such a way? Also, it might be
necessary to address other theoretical frameworks that argue otherwise. For
instance, the focus theory of normative conduct (Cialdini et al., 1990), the theory
of normative social behavior (Lapinski & Rimal, 2005), the reasoned action
approach (Fishbein, 2009) suggest that violation of injunctive norms does not
necessarily go along with violation of descriptive norms, and vice versa.

The authors commented that they see legal violations as a subset of the broader
category of norm violations. So, it looks like this research approach suggests that
violating the law also means violating social norms. To this point, please address
this argument from social norm theorists:

“Different from laws, norms are socially negotiated and contextually dependent modes
of conduct; laws are explicitly codified proscriptions that link violations with
their corresponding punitive measures. Laws are not socially negotiated (although
their enforcement might be), whereas norms and their transgressions, by definition,
are negotiated through social interaction. This is an important criterion because it
explains why the same mode of conduct (e.g., littering) is acceptable in one social
context (littered environment) but not in another (clean environment; Cialdini,
Reno, & Kallgren, 1990). Laws and norms can certainly reinforce each other. For
example, smokers may choose to refrain from lighting up in a public place for a
number of reasons, including legal (fear of being penalized) or normative (fear of
being accosted by someone in the vicinity), both of which lead to the same outcome
(not lighting up). At other times, the two may act in opposition to each other, as
when underage college students follow alcohol-drinking norms despite this behavior
being illegal.” (page 394, Rimal & Lapinski, 2015)

Berkowitz, A. D. (2005). An overview of the social norms approach. Changing the
culture of college drinking: A socially situated health communication campaign, 1,
193-214.

Chung, A., & Rimal, R. N. (2016). Social norms: A review. Review of Communication
Research, 4, 1-28.

Hue, D.T., Brennan, L., Parker, L. & Florian, M. (2015). But I am normal: Safe
driving in Vietnam. Journal of Social Marketing, 5(2), 105-124.

Lapinski, M. K., & Rimal, R. N. (2005). An explication of social norms.
Communication Theory, 15(2), 127-147.

Rimal, R. N., & Lapinski, M. K. (2015). A re-explication of social norms, ten
years later. Communication Theory, 25(4), 393-409.

Tankard, M. E., & Paluck, E. L. (2016). Norm perception as a vehicle for social
change. Social Issues and Policy Review, 10(1), 181-211.

2. This study focuses on the association between norm violation and perception of
power. The authors defined power as the perceived potential to influence others.
Additionally, they suggested that the perception of someone having the capacity to
do what that someone wants, it signals that the person has the capacity to influence
others (perception of power; line 58-60, page 4). Following this logic, a person who
does not buy a ticket is perceived as having the potential to influence others. I am
still confused with this logic. How is it possible that we travel on a bus and
witness a stranger not buying a ticket would make us think that that person has the
potential to influence us and others? In this scenario, I might think that the
person possesses some degrees of autonomy to conduct such a behavior. Yet, autonomy
is conceptually different from power and does not always lead to the attribution of
power. So, an inference from a high degree of autonomy to a high level of power
sounds like a leap in logic. The authors cited several studies to back up this
argument in their response, but the manuscript should speak for itself considering
that this is a pivotal theorization in this study.

Additionally, the authors wrote that “people who violate norms demonstrate that they
can behave as they wish.” How do we tell if people would attribute someone who does
not buy a ticket either as the person wishes to do so or that the person has no
choice at all? If the attribution is related to the second scenario, does that still
mean that the person is perceived to have the capacity to influence others? This
situation seems to relate to observers’ perceptions of efficacy of a norm violator
as well as observers’ attributions of the norm violator's traits. Attribution theory
suggests that human tends to attribute others’ negative behaviors as causally due to
internal factors and with less positive traits (e.g., fundamental attribution
errors). As such, a norm violator can be attributed with more negative attributes
(e.g., poor, desperate) than positive attributes (e.g., rich, high self-efficacy).
Isn’t it logical to think that positive attributes would be more likely to associate
with higher perception of power?

3. Operationalization of perceived power: It might be helpful to see the specific
items used to measure perceived power. Right now, the manuscript says that the
authors measured this construct by items like “I think this person has a great deal
of power,” which does not tell if participants understood that power was about the
potential ability to influence themselves and others. It would also be more
informative for reviewers and readers to see the specific items measuring other
scales because the items were adapted to this research situation.

4. What has been the common context of the studies the authors cited? Were these
studies mostly conducted in the western context where law and order and
transportation infrastructure are to some extent more stable than that in developing
countries? It is hard to fathom that a thieve on a public bus in a non-western
country (norm violator) would be perceived by on-lookers as having the potential to
influence others (power). It is also hard to think of an illegal drug user as being
someone who has power to influence others. I wonder if there is such a line of
research related to this study’s main theoretical framework to be able to be
generalized with a global implication. Even in the review of Stamkou et al. (2021)
that the authors cited, this norm violation – perceived power linkage was shown to
have contradicting effects in India. To this point, I still see that there’s a
significant challenge to persuade readers of the causal link between the observation
of norm-violation behaviors and perceived power.

5. The citation of perceived norm types should be acknowledged to Cialdini et al.
(1990) who coined the terms, which then became widely adopted in social science.

6. The term “costly behavior” should be clearly defined and with an example. Perhaps,
not all readers will have the in-depth knowledge of the authors’ research
discipline.

Reviewer #3: (No Response)

7. PLOS authors have the option to publish the peer
review history of their article (what does this mean?). If published, this will
include your full peer review and any attached files.

If you choose “no”, your identity will remain anonymous but your review may still be
made public.

**Do you want your identity to be public for this peer review?** For
information about this choice, including consent withdrawal, please see our
Privacy Policy.

Reviewer #1: No

Reviewer #2: No

Reviewer #3: No

---

## [Author Response · Author response to Decision Letter 1]

28 Jun 2021

Responses to Editor comments

Thank you for submitting your manuscript to PLOS ONE. After careful consideration, we
feel that it has merit but does not fully meet PLOS ONE’s publication criteria as it
currently stands. Therefore, we invite you to submit a revised version of the
manuscript that addresses the points raised during the review process.

Response: Thank you for giving us the opportunity to submit a second revised version
of our manuscript.

In the revised version of the paper, please try to clarify the aspects related to
descriptive and injunctive norms, how the perceived power has been measured, the
specificity / limitations of the study. When revising the paper, please consider the
reviewers' comments listed at the bottom of the email.

Response: We have carefully considered the comments of Reviewer 2 and clarified the
concepts of descriptive and injunctive norms, the measurement of perceived power,
and the limitations of the study.

Responses to Reviewer #2: 

I appreciate the authors’ efforts to address reviewers’ comments. 

Response: Thank you for your additional and helpful comments. 

1. Descriptive and injunctive norms were not clearly defined in the revised
manuscript. These norms are individually perceived. Social norms can be examined at
the collective level, which is different from social norm existing at the
individual/perceived level. At the collective level, both types of norms can
converge, but not necessarily so at the perceived level. People are not always
cognizant of the prevailing descriptive or injunctive norms in certain contexts
(please see Tankard & Paluck, 2016). The social norm approach, therefore,
suggests that misperception of social norms is a problematic issue for
norm-violating behaviors (please see Berkowitz, 2005).

Response: Thank you for your concrete suggestion for better defining descriptive and
injunctive norms. In the revised manuscript, we now state on page 3: “Injunctive and
descriptive norms are individually perceived but when people are cognizant of
prevailing norms and endorse these norms, both types of norms can converge and be
shared at the collective level [18]”. 

 Also, please note that in our studies, people did perceive the behavior of the norm
violator to be violating of norms (which is evident from the strong main effects on
the manipulation check; eta squared = .845 in S1, and eta squared = .715 in S2). So,
even though not everyone is always aware of the prevailing (descriptive or
injunctive) norms in a certain situation, the behavior we studied in our research
seems to be perceived consistently across participants. 

The authors wrote that injunctive norms and descriptive norms almost always work in
the same directions. They wrote that behaviors that are endorsed as appropriate by
the majority of the members of a group (injunctive norms) also tend to be enacted by
the majority of the members of a group (descriptive norms). However, there are many
situations where these two types of normative influences do not overlap, such as
when people approve of, but do not practice, particular behaviors (Cialdini et al.,
1990). Descriptive norms and injunctive norms can also be antagonistic, and they may
provide us with conflicting information about normative behaviors in a given context
(Lapinski & Rimal, 2005). For example, consider these norm-violating behaviors:
drinking, smoking, speeding, etc. (please see, for example, Chung & Rimal, 2016;
Hue et al., 2015).

The authors responded that “almost all prior research on responses to norm violators
has examined behaviors that would be considered violations of both descriptive and
injunctive norms.” Perhaps, this manuscript needs to speak for itself as to why
these two types of norms are almost all considered in such a way? Also, it might be
necessary to address other theoretical frameworks that argue otherwise. For
instance, the focus theory of normative conduct (Cialdini et al., 1990), the theory
of normative social behavior (Lapinski & Rimal, 2005), the reasoned action
approach (Fishbein, 2009) suggest that violation of injunctive norms does not
necessarily go along with violation of descriptive norms, and vice versa.

Response: We fully agree that there are situations in which descriptive and
injunctive norms do not overlap and that violation of injunctive norms does not
necessarily go along with violation of descriptive norms. Your comment led us to
realize that the claims we made about the frequent convergence of descriptive and
injunctive norms we provided in the previous version of the paper may have been too
strong, and we have therefore moderated our claims in the new revision. That said,
in our studies we explicitly used a scenario in which a formal norm (a contract
between the company operating the train and the passenger using its services) is
violated (not buying a train ticket) that is likely endorsed and enacted by most
members of a (western) society. In the revision, we have connected our claims about
convergence of descriptive and injunctive norms more tightly to this specific
operationalization so as not to imply that such convergence always occurs. To keep a
clear focus in our paper, we decided not to elaborate further on other theoretical
frameworks that address conflicting information about normative behaviors and
possible discrepancies between the violation of injunctive and descriptive norms.
Instead, in our discussion section we now reflect on our assumption that the
participants in our study were cognizant of the norm to buy a train ticket and
tended to endorse and enact this norm. The added information reads (page 24: “Third,
we considered the norm violation in this study as a violation of a descriptive and
injunctive legal norm. We assumed that buying a train ticket is a well-known legal
norm enacted and endorsed as appropriate by most study participants. Although we did
not test this assumption, the results of the manipulation checks in both studies
showed that participants perceived the behavior of the norm violator to be violating
of norms” and “Future research on norm violation could pay more attention to the
actual endorsement and enactment of specific norms among study participants.
Additionally, future research could examine situations where the violation of an
injunctive norm does not constitute a violation of a descriptive norm and vice versa
[43-44] to understand how participants differentially respond to violations of such
more complicated normative influences”.

The authors commented that they see legal violations as a subset of the broader
category of norm violations. So, it looks like this research approach suggests that
violating the law also means violating social norms. To this point, please address
this argument from social norm theorists:

“Different from laws, norms are socially negotiated and contextually dependent modes
of conduct; laws are explicitly codified proscriptions that link violations with
their corresponding punitive measures. Laws are not socially negotiated (although
their enforcement might be), whereas norms and their transgressions, by definition,
are negotiated through social interaction. This is an important criterion because it
explains why the same mode of conduct (e.g., littering) is acceptable in one social
context (littered environment) but not in another (clean environment; Cialdini,
Reno, & Kallgren, 1990). Laws and norms can certainly reinforce each other. For
example, smokers may choose to refrain from lighting up in a public place for a
number of reasons, including legal (fear of being penalized) or normative (fear of
being accosted by someone in the vicinity), both of which lead to the same outcome
(not lighting up). At other times, the two may act in opposition to each other, as
when underage college students follow alcohol-drinking norms despite this behavior
being illegal.” (page 394, Rimal & Lapinski, 2015)

Response: Again, thank you for your concrete input. We now address the difference
between laws and norms in the Discussion section. Our text on page 24 reads as
follows: “Train passengers who do not buy a train ticket transgress a legal norm and
run the risk of being formally penalized by means of a fine. Note that laws, as
opposed to social norms, are not negotiated through social interaction, which means
that people’s responses to violating the legal norm to buy a train ticket may be
relatively similar across social contexts [42]. Prior research has shown that legal
norm violations such as financial fraud [5] or illegal parking [23] elicit similar
responses from observers as non-legal norm violations such as arriving late to a
meeting [8] or putting one’s feet on another’s table [5]”.

2. This study focuses on the association between norm violation and perception of
power. The authors defined power as the perceived potential to influence others.
Additionally, they suggested that the perception of someone having the capacity to
do what that someone wants, it signals that the person has the capacity to influence
others (perception of power; line 58-60, page 4). Following this logic, a person who
does not buy a ticket is perceived as having the potential to influence others. I am
still confused with this logic. How is it possible that we travel on a bus and
witness a stranger not buying a ticket would make us think that that person has the
potential to influence us and others? In this scenario, I might think that the
person possesses some degrees of autonomy to conduct such a behavior. Yet, autonomy
is conceptually different from power and does not always lead to the attribution of
power. So, an inference from a high degree of autonomy to a high level of power
sounds like a leap in logic. The authors cited several studies to back up this
argument in their response, but the manuscript should speak for itself considering
that this is a pivotal theorization in this study.

Additionally, the authors wrote that “people who violate norms demonstrate that they
can behave as they wish.” How do we tell if people would attribute someone who does
not buy a ticket either as the person wishes to do so or that the person has no
choice at all? If the attribution is related to the second scenario, does that still
mean that the person is perceived to have the capacity to influence others? This
situation seems to relate to observers’ perceptions of efficacy of a norm violator
as well as observers’ attributions of the norm violator's traits. Attribution theory
suggests that human tends to attribute others’ negative behaviors as causally due to
internal factors and with less positive traits (e.g., fundamental attribution
errors). As such, a norm violator can be attributed with more negative attributes
(e.g., poor, desperate) than positive attributes (e.g., rich, high self-efficacy).
Isn’t it logical to think that positive attributes would be more likely to associate
with higher perception of power?

Response: In order to better explain the link between norm violation and perceptions
of power, we have revised the text on pages 3 and 4. The text on page 3 now reads as
follows: “Social norms – implicit or explicit rules or principles that are
understood by members of a group and that guide and/or constrain behavior [1] –
create a shared understanding of what is acceptable within a given context and
thereby contribute to the functioning of social collectives [2-4]. Accordingly,
research has documented that people who violate norms tend to elicit negative
responses in others, including unfavorable social perceptions [5], negative emotions
[6- 8], scolding [9], gossip [10], and punishment [11-13]. Intriguingly, however,
research has also demonstrated that norm violators are perceived as powerful [5],
high in status [14], and influential [15]”. 

The text on pages 3-5 now reads as: “By ignoring the norms that bind others, norm
violators demonstrate that they can act as they wish and do not fear interference
from others [5]. This is a freedom that typically comes with higher rank [19]. The
influential approach/inhibition theory of power [20] states that power, which is
commonly defined as asymmetrical control over valuable resources that enables
influence, liberates behavior, whereas powerlessness constrains it. Indeed, ample
research supports that power renders people more likely to act, even if the
resulting behavior is inappropriate or harmful [21-22]. Because behavioral freedom
is thus intimately associated with power, people who observe unchecked behavior of
others may make inferences about others’ level of power. Indeed, people who act as
they wish and disregard social norms are perceived as having high status [14],
influence [15], and power [5]. Furthermore, these perceptions can, under particular
circumstances, fuel actual granting of power, for instance via the conferral of
control over outcomes, voting, and leadership endorsement [23-24]. In line with the
notion that power liberates behavior, previous research has demonstrated that norm
violators are perceived as powerful because they appear to experience the freedom to
act as they please [14-15, 5] – that is, they are high on volitional capacity. In
other words, norm violators are perceived as powerful because their behavior signals
an underlying quality, namely the freedom to act at will. This argument resonates
with costly signaling theory [25-26], which states that any seemingly costly
behavior (involving large investments or risks of receiving negative outcomes)
functions as a signal of an underlying characteristic [25-26]. An example of costly
behavior is the reckless driving of young men as to show their strength and skills
to peers and potential mates, risking serious injury or death – a type of behavior
that is under particular circumstances “rewarded” with power [27]. Norm violations
are potentially costly as they are frequently sanctioned [14] by means of formal
(e.g., legal) punishment [28] and/or informal (social) punishment (e.g., anger,
social exclusion [29-30]). According to costly signaling theory, people who engage
in potentially costly norm-violating behavior signal that they possess traits that
allow them not to worry about interferences from others. Because this capacity to do
what one wants is typically reserved for the powerful [31], norm violators appear
powerful when there are no additional cues that provide direct information about
this attribute [5]”. 

Thank you for your thoughts about other (than power) perceptions of norm violators.
Your intuition that norm violators are generally perceived negatively is borne out
by previous research, which we believe makes it all the more interesting that people
still perceive norm violators as powerful – except when they are sanctioned, as we
demonstrate in the current paper, because sanctioning severs the link between
perceived volitional capacity and perceived power. In the introduction section on
page 3, we now briefly discuss previous work that has documented negative responses
to norm violations to better contextualize the current findings and enable nuanced
conclusions. Your comment also led us to think about associations between perceived
power and other social perceptions of norm violators more broadly, which we will
seriously consider when preparing future research on responses to norm
violations.

3. Operationalization of perceived power: It might be helpful to see the specific
items used to measure perceived power. Right now, the manuscript says that the
authors measured this construct by items like “I think this person has a great deal
of power,” which does not tell if participants understood that power was about the
potential ability to influence themselves and others. It would also be more
informative for reviewers and readers to see the specific items measuring other
scales because the items were adapted to this research situation.

Response: In the method section of Study 1, we refer to the supplementary material
containing the scale items. 

Perceived Power (Anderson & Galinsky, 2006) was measured with the following
items: 

1. He can get people to listen to what he says. 

2. His wishes do not carry much weight. [reverse scored] 

3. He can get others to do what he wants. 

4. Even if he voices them, his views have little sway. [reverse scored] 

5. He thinks he has a great deal of power. 

6. His ideas and opinions are often ignored. [reverse scored] 

7. Even when he tries, he is not able to get his way. [reverse scored] 

8. If he wants to, he gets to make the decisions. 

In addition, we have added one more sample item of the perceived power scale in the
Method section of Study 1.

4. What has been the common context of the studies the authors cited? Were these
studies mostly conducted in the western context where law and order and
transportation infrastructure are to some extent more stable than that in developing
countries? It is hard to fathom that a thieve on a public bus in a non-western
country (norm violator) would be perceived by on-lookers as having the potential to
influence others (power). It is also hard to think of an illegal drug user as being
someone who has power to influence others. I wonder if there is such a line of
research related to this study’s main theoretical framework to be able to be
generalized with a global implication. Even in the review of Stamkou et al. (2021)
that the authors cited, this norm violation – perceived power linkage was shown to
have contradicting effects in India. To this point, I still see that there’s a
significant challenge to persuade readers of the causal link between the observation
of norm-violation behaviors and perceived power.

Response: With regard to the question of whether a thieve on a bus would be perceived
as being capable of influencing others, we believe the answer is a clear yes.
Interpersonal influence stems not only from admirable qualities such as competence,
expertise, and skill (which are related to prestige) but also from attributes such
as assertiveness, intimidation, and coercion (which are related to dominance); see,
for instance, Anderson and Kilduff (2009) and Cheng et al. (2013). With regard to
culture, the studies we cite were mostly conducted in a western context,
consequently showing a link between norm violation and observers’ power perceptions.
Stamkou et al.’s (2019) cross-cultural comparison of responses to norm violations
revealed that the link between norm violation and power perceptions is positive in
individualistic cultures, but negative in collectivistic cultures (Stamkou et al.,
2019). Moreover, individuals in tighter cultures are less willing to endorse norm
violators as leaders, compared to those in looser cultures. It is clear from this
cross-cultural study that observers’ responses to norm violations are indeed
influenced by the cultural context in which the violation occurs. We have addressed
this issue in the final paragraph of the Discussion section on page 25.

5. The citation of perceived norm types should be acknowledged to Cialdini et al.
(1990) who coined the terms, which then became widely adopted in social science.

Response: Thank you for noting this omission. We have included the reference to
Cialdini et al. (1990).

6. The term “costly behavior” should be clearly defined and with an example. Perhaps,
not all readers will have the in-depth knowledge of the authors’ research
discipline.

Response: We have revised the text on costly behavior. The text on page 5 now reads
as: “This argument resonates with costly signaling theory [25-26], which states that
any seemingly costly behavior (involving large investments or risks of receiving
negative outcomes) functions as a signal of an underlying characteristic [25-26]. An
example of costly behavior is the reckless driving of young men as to show their
strength and skills to peers and potential mates, risking serious injury or death –
a type of behavior that is under particular circumstances “rewarded” with power
[27].”

References not included in the paper:

Cheng, J. T., Tracy, J. L., Foulsham, T., Kingstone, A., & Henrich, J. (2013).
Two ways to the top: Evidence that dominance and prestige are distinct yet viable
avenues to social rank and influence. Journal of Personality and Social Psychology,
104, 103-125.

References added to the paper:

1. Cialdini RB, Trost MR. Social influence: Social norms, conformity, and compliance.
In Gilbert DT, Fiske ST, Lindzey G, editors. Handbook of social psychology. New
York, NY: McGraw-Hill; 1998. pp. 151–192.

2. Jetten J, Hornsey MJ. Deviance and dissent in groups. Annu Rev Psychol. 2014; 65:
461–485. doi: 10.1146/annurev-psych-010213-115151.

3. Tomasello M, Vaish A. Origins of human cooperation and morality. Annu Rev Psychol.
2013; 64: 231–255. doi: 10.1146/annurev-psych-113011-143812.

6. Gutierrez R, Giner-Sorolla R. Anger, disgust, and presumption of harm as reactions
to taboo-breaking behaviors. Emotion. 2007; 7(4): 853–868. doi:
10.1037/1528-3542.7.4.853 853.

7. Ohbuchi KI, Tamura T, Quigley BM, Tedeschi JT, Madi N, Bond MH, Mummendey A.
Anger, blame, and dimensions of perceived norm violations: Culture, gender, and
relationships. J Appl Soc Psychol. 2004; 34(8): 1587–1603. doi: 10
.1111/j.1559-1816.2004.tb02788.x.

9. Vaish A, Missana M, Tomasello M. Three-year-old children intervene in third-party
moral transgressions. Br J Dev Psychol. 2011; 29(1): 124–130. doi:
10.1348/026151010X532888.

10. Beersma B, Van Kleef GA. Why people gossip: An empirical analysis of social
motives, antecedents, and consequences. J Appl Soc Psychol. 2012; 42(11): 2640–2670.
doi: 10.1111/j.1559-1816.2012.00956.x.

11. Fehr E, Fischbacher U. Third-party punishment and social norms. Evol Hum Behav.
2004; 25(2): 63–87. doi: 10.1016/S1090-5138(04)00005-4.

12. Marques JM, Abrams D, Serôdio RG. Being better by being right: Subjective group
dynamics and derogation of ingroup deviants when generic norms are undermined. J
Pers Soc Psychol. 2001; 81(3): 436–447. doi: 10.1037/0022-3514.81.3.436.

13. Yamagishi T. The provision of a sanctioning system as a public good. J Pers Soc
Psychol. 1986; 51(1): 110–116. doi: 10.1037/0022-3514.51.1.110.

15. Stamkou E, Van Kleef GA, Homan AC. The art of influence: When and why deviant
artists gain impact. J Pers Soc Psychol. 2018; 115(2): 276-303. doi:
10.1037/pspi0000131.

18. Tankard ME, Paluck EL. Norm perception as a vehicle for social change. Soc Iss
Policy Rev. 2016; 10(1): 181-211. doi: 10.1111/sipr.12022.

19. Galinsky AD, Magee JC, Gruenfeld DH, Whitson JA, Liljenquist KA. Power reduces
the press of the situation: Implications for creativity, conformity, and dissonance.
J Pers Soc Psychol. 2008; 95(6): 1450–1466. doi: 10.1037/a0012633.

20. Keltner D, Gruenfeld DH Anderson C. Power, approach, and inhibition. Psychol Rev,
2003; 110(2): 265-284. doi: 10.1037/0033-295X.110.2.265.

21. Cho M, Keltner D. Power, approach, and inhibition: Empirical advances of a
theory. Cur Opin Psychol. 2020; 33: 196-200. doi: 10.1016/ j.copsyc.2019.08.013.

22. Pike BE, Galinsky AD. Power leads to action because it releases the psychological
brakes on action. Cur Opin Psychol. 2020; 33: 91-94. doi:
10.1016/J.COPSYC.2019.06.028.

27. Van Kleef GA, Heerdink MW, Cheshin A, Stamkou E, Wanders F, Koning LF, et al. No
guts, no glory? How risk-taking shapes dominance, prestige, and leadership
endorsement. J Appl Psychol. (in press).

42. Rimal RN, Lapinski MK. A re-explication of social norms, ten years later. Commun
Theor. 2015; 25(4): 393-409. doi: 10.1111/comt.12080.

43. Cialdini RB, Reno RR, Kallgren CA. A focus theory of normative conduct: Recycling
the concept of norms to reduce littering in public places. J Pers Soc Psychol. 1990;
58(6): 1015-1026. doi: 10.1037/0022-3514.58.6.1015.

44. Lapinski MK, Rimal RN. An explication of social norms. Commun Theor. 2005; 15(2):
127-147. doi:10.1111/j.1468-2885.2005.tb00329.x

to reviewers R2 .docx
---

## [Editor Report · Decision Letter 2]

30 Jun 2021

How norm violators rise and fall in the eyes of others: The role of sanctions

PONE-D-20-25007R2

Dear Dr. van Vianen,

We’re pleased to inform you that your manuscript has been judged scientifically
suitable for publication and will be formally accepted for publication once it meets
all outstanding technical requirements.

Kind regards,

Camelia Delcea

Academic Editor

PLOS ONE
---

## [Editor Report · Acceptance letter]

21 Jul 2021

PONE-D-20-25007R2 

How norm violators rise and fall in the eyes of others: The role of sanctions 

Dear Dr. van Vianen:

I'm pleased to inform you that your manuscript has been deemed suitable for
publication in PLOS ONE. Congratulations! Your manuscript is now with our production
department. 

Kind regards, 

on behalf of

Dr. Camelia Delcea 

Academic Editor

PLOS ONE